# miRNAs and Substances Abuse: Clinical and Forensic Pathological Implications: A Systematic Review

**DOI:** 10.3390/ijms242317122

**Published:** 2023-12-04

**Authors:** Carla Occhipinti, Raffaele La Russa, Naomi Iacoponi, Julia Lazzari, Andrea Costantino, Nicola Di Fazio, Fabio Del Duca, Aniello Maiese, Vittorio Fineschi

**Affiliations:** 1Department of Surgical Pathology, Medical, Molecular and Critical Area, Institute of Legal Medicine, University of Pisa, 56126 Pisa, Italy; c.occhipinti4@studenti.unipi.it (C.O.); n.iacoponi2@studenti.unipi.it (N.I.); j.lazzari@studenti.unipi.it (J.L.); a.costantino8@studenti.unipi.it (A.C.); 2Department of Clinical Medicine, Public Health, Life Sciences, and Environmental Sciences, University of L’Aquila, 67100 L’Aquila, Italy; raffaele.larussa@univaq.it; 3Department of Anatomical, Histological, Forensic and Orthopaedic Sciences, Sapienza University of Rome, Viale Regina Elena 336, 00161 Rome, Italy; nicola.difazio@uniroma1.it (N.D.F.); fabio.delduca@uniroma1.it (F.D.D.); vittorio.fineschi@uniroma1.it (V.F.)

**Keywords:** substance abuse, addiction, alcohol addiction, cocaine addiction, opioids addiction, methamphetamine addiction, miRNA, microRNA

## Abstract

Substance addiction is a chronic and relapsing brain disorder characterized by compulsive seeking and continued substance use, despite adverse consequences. The high prevalence and social burden of addiction are indisputable; however, the available intervention is insufficient. The modulation of gene expression and aberrant adaptation of neural networks are attributed to the changes in brain functions under repeated exposure to addictive substances. Considerable studies have demonstrated that miRNAs are strong modulators of post-transcriptional gene expression in substance addiction. The emerging role of microRNA (miRNA) provides new insights into many biological and pathological processes in the central nervous system: their variable expression in different regions of the brain and tissues may play a key role in regulating the pathophysiological events of addiction. This work provides an overview of the current literature on miRNAs involved in addiction, evaluating their impaired expression and regulatory role in neuroadaptation and synaptic plasticity. Clinical implications of such modulatory capacities will be estimated. Specifically, it will evaluate the potential diagnostic role of miRNAs in the various stages of drug and substance addiction. Future perspectives about miRNAs as potential novel therapeutic targets for substance addiction and abuse will also be provided.

## 1. Introduction

Drug addiction is a significant public health issue worldwide. The term drug addiction comprehends a series of actions culminating in loss of control over drug intake, craving, and withdrawal. Another expression used in this regard is substance use disorder (SUD), and it refers to different types of behaviors varying from sporadic use of substances of abuse, abuse, dependence, or addiction.

The Diagnostic and Statistical Manual of Mental Disorders (DSM) provides a differential clinical diagnosis of these behaviors [1]. Repeated exposure to drugs of abuse frequently leads to a change in feeling, alarming behavior, temporary satiation, loss of control and negative consequences, and clear symptoms of addiction [2]. The evolution from casual drug use to compulsive drug seeking seems to be strictly related to the reward and motivation brain circuitry [3], an adaptive response to long-term drug use. One of the hallmarks of addiction is represented by the change from a positive reward, usually called a “high”, to negative stimuli during abstinence. Different brain regions are involved in these processes. In particular, acute rewarding effects are connected to an increase in dopamine release in the Nucleus Accumbens (NAc), while long-term plasticity in corticostriatal circuits are linked to compulsive drug seeking and susceptibility to relapse [4].

This behavior can be analyzed in animal models through Pavlovian-style conditioning such as operant conditioned lever-pressing self-administration (SA) or conditioned placement preference (CPP). Additionally, according to the Centers for Disease Control and Prevention, in the United States, deaths from drug overdoses topped 100,000 in 2022 [5].

There is, however, a lack of understanding of the underlying molecular mechanisms of addiction. Drug screening and confirmation of dependence or use disorder are mainly conducted by analyzing drugs and their metabolites in blood samples, which are not stable with levels constantly decreasing [6]. Furthermore, diagnoses of the severity of drug use disorders are mainly based on subjective reports and there are no objective biomarkers available. Therefore, the identification of stable and objective biomarkers for drug abuse is crucial.

Indeed, epigenetic, and social factors are involved in the development of addiction. On one hand, social and psychological factors predict a drug abuser profile; on the other hand, molecular processes are involved [7,8]. Although clinical studies have previously shown the impact of social factors on drug use [9], it is only recently that neuroscience has devoted careful attention to examining the influence of social factors on drug use and addiction in animal models [10]. Further specific attention should be given to the development of new disorders triggered by adaptive responses to drug use and those relapsed thereafter [11,12,13,14].

It is recognized that drugs of abuse operate by modulating gene expression and producing their rewarding effects of euphoria or pleasure through the interaction with the mesolimbic dopaminergic system, leading to persistent alterations in the reward-related and memory-related brain centers. These interactions translate into the alteration of proteins and gene expression by regulating transcription factors, chromatin structures, and small non-coding RNA (ncRNA) [2]. One of the most studied classes of ncRNA in drug addiction is that of microRNAs (miRNA) [15].

miRNAs are small non-coding RNA transcripts, comprising 19–25 ribonucleotides, that are not translated into proteins but are capable of regulating gene expression at a post-transcriptional level [16]. They have the ability to destabilize and cleave mRNA, therefore actively regulating their transcription. Moreover, the silencing effect of miRNAs on their target gene expression appears to be highly specific allowing a precise regulation of physiological responses [15]. Given their role in epigenetic regulation and synaptic plasticity, miRNAs have the potential to be used as markers of drug abuse both on brain tissue samples, but also on blood vessels and bodily fluids such as blood and saliva. By understanding which phases of addiction are regulated by specific miRNAs and which miRNAs are altered by different types of drugs, new tailored and ultra-selective therapies can be developed to aid in the battle against drug addiction. For this reason, a review of the currently available data on miRNAs involved in the main classes of drugs of abuse (alcohol, cocaine, methamphetamine, and opiates) has been carried out.

## 2. Materials and Methods

### 2.1. Eligibility Criteria

The present systematic review was carried out according to the Preferred Reporting Items for Systematic Review (PRISMA) standards [17]. We used an evidence-based model for framing a PICO.

### 2.2. Search Criteria and Critical Appraisal

A systematic literature search and a critical appraisal of the collected studies were conducted. An electronic search of PubMed, Science Direct Scopus, and Excerpta Medica Database (EMBASE) from the inception of these databases to September 2023 was performed. The search terms were “miRNA expression in addiction”; “miRNAs during alcohol addiction”; “miRNAs during cocaine addiction”; “miRNAs during methamphetamine addiction”; “miRNAs during opioids addiction” in the title, abstract, and keywords. The bibliographies of all identified documents were reviewed and cross-checked for the other relevant literature. Methodological assessment of each study was conducted according to PRISMA standards, including assessment of bias. Data collection included study selection and data extraction. Three researchers (C.O., J.L, N.I.) independently reviewed those documents whose title or abstract seemed relevant and selected those that analyzed miRNAs. The question of the suitability of miRNAs was resolved by consensus among the researchers. The unpublished or grey literature was not searched. The extraction of data was performed by three investigators (E.T., A.M., A.C.) and reviewed by two other researchers (V.F., P.F.). Only English-language papers or abstracts were included in the search.

### 2.3. Search Results and Included Studies

An appraisal based on titles and abstracts, as well as a hand search of reference lists, were carried out. The reference lists of all located articles were reviewed to detect the still unidentified literature. The resulting reference lists were then screened for title and abstract, leaving 216 articles, for further consideration, that were then screened based on their abstract to identify their relevance in respect to the following: Estimate diagnosis process; Clinical features analyzed; Circumstantial data evaluation; Study design.

The methodology of our search strategy is represented in Figure 1.

A further categorization of the articles included in this study was made based on the main parameter under study, as shown in the following table (Figure 1).

Non-English articles were excluded. The inclusion criteria were as follows: (1) original research articles; (2) case reports/series. Reviews and meta-analyses were excluded. Studies conducted on cultured cells, animals (mainly rats and mice), and humans that described the role of miRNAs in different stages of addiction on four types of substances of abuse (alcohol, cocaine, opioids, and methamphetamine) were selected. Among them, studies measuring miRNAs in CNS and some body fluids (mainly blood and saliva) from animals and humans were included, as also were studies with a control group. These publications were carefully assessed, considering the main objectives of this review. After this evaluation, 132 scientific papers remained.

### 2.4. Risk of Bias

This systematic review has several strengths that include the amount and breadth of the studies, which span the globe; the hand search and scan of reference lists for the identification of all relevant and significant studies; and a flowchart that describes in detail the study selection process. Included studies were evaluated according to the Quality Assessment of Diagnostic Accuracy Studies 2 (QUADAS-2) tool to assess the risk of bias. This review includes studies that were published in a time frame of 13 years (from 2010 to 2023); thus, despite our efforts to fairly evaluate the existing literature, study results should be interpreted taking into account that the accuracy of the clinical procedures, here reported, has changed over the years.

## 3. Results

### 3.1. miRNAs Expression in Alcohol Abuse

Alcohol is a neuro-depressant substance, able to determine the occurrence of short and long-term adverse effects. An unregulated consumption of alcohol may lead to a condition named “alcohol abuse”: it represents a spectrum of different unhealthy behaviors related to alcohol intake, ranging from binge drinking up to alcohol withdrawal, that comprises also entities such as alcohol use disorder (AUD) and alcohol dependence (AD). It has been widely demonstrated how alcohol abuse, in its various stages, regulates and is in turn regulated by miRNAs expression (see Table 1). Numerous are the miRNAs regulated by the consumption of alcohol, just as numerous are the targets on which they act. In a work by Lewohl and colleagues, it emerged that a typical pattern of interaction between miRNAs and their targets provides for an inverse correlation; it is usual to have an upregulation of miRNAs alongside a downregulation of their targets, or vice versa [18].

One of the most targeted molecules by miRNAs is the endogenous protective factor BDNF, which is usually found in the human brain. It works as a regulator of the transcriptional and translational mechanism when it is activated through the TrkB receptor. The BDNF/TrkB pathway has a key role in the development of the central nervous system (CNS) as it is involved in the formation of synapses, differentiation of neural cells, and synaptic plasticity.

Ehinger et al. [19] found out that moderate use of alcohol determines an increase of BDNF levels in the dorsolateral striatum (DLS), thus activating the TrkB/ERK signaling and inducing the dopamine D3 receptor, so as to maintain a moderate intake of the substance. Moreover, they also discovered that heavy use of alcohol is instead associated with a reduction of BDNF levels in the prefrontal cortex (PFC) and in serum 8 weeks after the intake of the substance, suggesting that after a past heavy alcohol intake, low BDNF protein levels may be measured.

In the study, it was found that BDNF is the target of numerous miRNAs, such as miR30a-5p, miR195, miR191, and miR206, whose levels are consistently increased after alcohol exposure even in serum. They hypothesized that such miRNAs are firstly produced in mPFC, then transported in the blood, thus acting as potential diagnostic tools in order to understand alcohol’s role in the brain. Even other studies reported similar results [2,4,43,44,45].

In addition, some of those studies further found out that miR-195 and miR-30a-5p are also upregulated after episodes of binge drinking and after one day of abstinence, whereas miR206 was upregulated even after three weeks of alcohol exposure, during the phase of withdrawal [20,22].

According to the studies conducted by Gowen [2], Heyer [4], Zhang [45], and Zhao [43], it seems that BDNF is also regulated by miR124a, which appears to have a key role in the mechanism of addiction to substances. In particular, miR124a levels were decreased in the DLS of rats after 15 days of ethanol intaking, and such featuring was associated with increased levels of BDNF [21]. On a practical level, the reduction of miR124a expression was related to a decrease of alcohol intake and, as a consequence, they suggest that it may be used to counteract vulnerability to alcohol dependence. Moreover, Mizuo et al. [26] found out that miR124 levels, along with miR132, remained upregulated 12 h after alcohol assumption, and also in the phase of abstinence, which means that it may play a significant role even in the relapse phase.

Another miRNA whose expression is modified subsequently to alcohol exposure is miR9. Several works highlighted how its levels tend to increase in the cells of the hypophysis, the supraoptic nucleus, and the striatum following acute intake of alcohol [4,25,43,45]. The target of miR9 is the BK channel, an ionic channel whose expression is usually reduced by alcohol.

By binding the BK channel, miR9 induces post-transcriptional modifications resulting in the downregulation of a specific subunit therefore making the BK channel less sensitive to alcohol. As a consequence, miR9 can determine a long-term adaptation that induces an increase in tolerance to ethanol [23].

Gu et al. [24] also identified HDA5 as a target of miR9. It is part of the histone deacetylases family and is negatively regulated by miR9. Its decreased expression in primary cortical neurons seemed to be responsible for behavioral responses to chronic drug exposure.

A further target of miR9 is the dopaminergic receptor DR2, where its lower levels have been associated with alcohol abuse. Such a result suggests that miR9 also affects the alcohol-related reward effect and can take part in the development of addiction 27 [25]. Even DR1 represents a target of miRNAs following alcohol exposure. In particular, this dopaminergic receptor is targeted by miR-382: miR-382 is downregulated after repeated alcohol administration whereas DR1 expression is increased. It was further demonstrated that inducing expression of miR-382 determines inhibition of DR1 and, consequently, reduction of alcohol consumption [27].

Moreover, overexpression of let-7d determines the downregulation of DR3 in the nucleus accumbens of rats, thus reducing alcohol consumption [21].

According to Most’s work, it seems that miRNAs affect and regulate other neurotransmission systems, like the glutamate one: for example, one of the targets of miR-411 is GluR2. miR-411 is reduced after chronic alcohol intake and knockdown of miR-411 reduces alcohol intake besides inducing expression of GluR2 [46].

Another circuit influenced by alcohol and miRNA expression is, clearly, the GABAergic one. Bekdash et al. [38] observed that during alcohol withdrawal, subunit Gabra4 of the GABA_A_ receptor was down-expressed. They demonstrated that knockdown of Gabra4 was inducted by transfection of miR-186, miR-24, miR-375, and miR-155 mimics. Conversely, transfection of inhibitors of such miRNAs determined restoration of the expression of Gabra4, thus suggesting that Gabra4 represents an additional target.

The serotoninergic circuit is also influenced by miRNA expression. Another receptor targeted by miRNAs is the glucocorticoid one (GR). It was observed that long-term alcohol exposure determined a reduction in GRα’s expression in the presence of raised levels of miR-124. Such a phenomenon may be implicated in alcohol addiction [31].

Interestingly, some studies focused on the capacity of miRNAs regulated by alcohol exposure, in controlling apoptotic processes. Among these miRNAs, there are miR21 and miR335, which act as functional antagonists [35], and miR29b, which exerts a protective role against ethanol-induced neurotoxicity [36].

MiRNAs are also expressed in synapses and their regulation can be altered by chronic alcohol consumption. In their experiments, conducted on murine models, Most et al. demonstrated that alcohol actually alters miRNA expression in glutamate synapses. Some of those miRNAs, such as miR-203, miR-18 a, and miR-374, were also identified in human brains [28].

miRNAs altered by alcohol intake could also be found within some microvesicles released by glial cells. This is the case of let-7b, whose target is represented by TLR7 and activates the nuclear factor kβ (NF-kβ), thus determining neurodegeneration. Some studies evidenced that in alcoholic humans and rats, there were increased levels of TLR7. Such a result was the effect of the increased release of let-7b in microvesicles, that contributed to ethanol-induced neurotoxicity mediated through TLR7 [29]

An interesting work conducted by Asquith et al. demonstrated that miR-181 and miR-221 were overexpressed not only in the brain’s tissues but also in the peripheral blood mononuclear cells (PBMC) following chronic assumption of ethanol in those cells [37].

Choi et al. [39] evaluated the sex-dependent effects of alcohol on miRNA expression. They found that different patterns of miRNAs were up or downregulated depending on whether they were evaluated in males or females. Even neurogenesis was differently affected by alcohol depending on sex: this was evaluated by measuring DCX, demonstrating that males were more sensitive to alcohol than females.

Whereas most of the studies are based on experiments on rats, Lim et al.’s results are based on studies conducted on postmortem brain tissue collected from humans affected by AUD. They examined eight different brain regions, where the region with the greatest number of miRNAs associated with AUD was the PFC. The main targets of the different miRNAs were represented by CREB signaling, IL-8 signaling, and Axonal Guidance Signaling, thus suggesting the involvement of miRNAs in regulating anxiety-like and excessive alcohol-drinking behaviors [32]. Moreover, Liu and Zhang demonstrated, on postmortem brain tissue, that chronic alcohol consumption may determine hypermethylation of both miRNAs and mRNAs in the region of NAc [47].

Other studies conducted on humans by Manzardo et al. [42] examined miRNA expression in the medial frontal cortex of alcoholic subjects. They found that twelve miRNAs were upregulated (including miR-3065-5p, miR-299-5p, miR-767-5p, miR-375, miR-29b, miR-377, miR-399) and two were downregulated (miR-572 and miR-3162). These miRNAs targeted genes implicated in cellular adhesion (as THBS2), differentiation of tissues (as CHN2), migration (as NDE1), and maturation of neurons and oligodendrocytes.

Furthermore, it has been demonstrated that miRNAs could also cross the placenta and potentially exert some teratogenic effects. It was observed that the expression of some of these miRNAs is influenced by exposition to alcohol in a way that they could be possibly used as biomarkers for forecasting the effects on infants with prenatal alcohol exposure [30].

Similar results were found by Tseng et al. who focused attention on miR-140, a molecule contained in microvesicles released from fetal neural stem cells, whose levels were increased after 3 days of alcohol exposure [34]. Other miRNAs that have been observed in the serum of mothers with alcohol abuse were miR122, miR126, miR216b, miR221, miR3119, miR3942, miR4704, miR4743, miR514, and miR602.

Finally, an interesting work was presented by Rosato et al. [48] Through an experiment conducted on 110 participants, they demonstrated that miRNAs measured in saliva could be used as biomarkers in order to obtain a non-invasive prediction for alcohol dependence. In particular, they identified a cluster of five salivary miRNAs with an accuracy for prediction of over 70%.

### 3.2. miRNAs Expression in Cocaine Abuse

Cocaine is one of the most consumed stimulants in the world. It is a natural sympathomimetic tropane alkaloid that can be extracted from the leaves of Erythroxylon coca, by either chewing the leaves or brewing tea. In the mid-1800s, cocaine was considered safe and used in toothache drops, nausea pills, energy tonics, and also, in the original recipe of the ‘Coca-Cola’ beverage. Nowadays, cocaine is usually found in two forms: cocaine hydrochloride, commonly known as ‘coke’, ‘blow’, or ‘snow’, a fine crystalline white powder, soluble in water and mainly consumed intranasally (‘sniffing’ or ‘snorting’), orally, or intravenously; and as ‘crack’ cocaine, a free base resulting from the reaction of cocaine hydrochloride and ammonium or baking soda, and typically consumed via inhalation. Cocaine has singular pharmacodynamic properties that allow its use both as a sympathomimetic stimulant of the central nervous system and as a local anesthetic.

The sympathomimetic and psychoactive effects are linked to the blockade of presynaptic transporters responsible for the reuptake of serotonin, noradrenaline, and dopamine. The anesthetic properties are related, on the other hand, to its capacity to block voltage-gated sodium channels through their stabilization in an inactive state. When cocaine is consumed, dopaminergic activity along the mesocorticolimbic pathways is intensified. This substance induces euphoria, improves concentration and alertness, increases libido, promotes a general sensation of well-being, and reduces appetite and fatigue. Nonetheless, its use can also determine insomnia, irritability, anxiety, impulsive behavior, and dysphoria, particularly after continued drug use. Moreover, it is correlated to cardiovascular effects such as tachycardia, vasoconstriction, and hypertension. Cocaine has an elevated abuse potential, by increasing dopamine concentration in the nucleus accumbens, the dorsal caudate nucleus, and the striatum [49]. MiRNAs expression in cocaine abuse are shown in Table 2.

Barreto-Valer et al. [50] evaluated the effects of cocaine on the expression of dopamine receptors and miR-133b. They demonstrated that cocaine alters dopaminergic receptor expression in various ways depending on both the receptor and the embryo’s developmental stage. Cocaine reduced the expression of miR-133 at 24 and 48 hpf and, conversely, increased pitx3 at 24 hpf. They then focused on the pitx3 target genes TH, DAT, and DRD2 expression, showing they were upregulated at 24 hpf. TH and DAT decreased at 48 hpf, whereas DRD2A increased and DRD2B was downregulated at both 24 and 48 hpf. Finally, cocaine decreased the expression of miR-133b in the central nervous system and at the periphery, particularly on somites. Therefore, the authors hypothesized that it could be involved in skeletal muscle development.

Bastle et al. [51] evaluated the direct binding of miR-495 to target genes showing that it significantly reduced the activity of BDNF, Camk2a, and Arc in vitro. NAc miR-495 was downregulated between 1 and 4 h post-injection in a region-specific manner. Moreover, BDNF and Camk2a were upregulated 2 h after cocaine injection in the NAc.

Cabana-Dominguez et al. [52] found that miR-9-5p, miR-101-3p, miR-124-3p, miR-124-5p, miR-137, miR-153-3p, and miR- 369-3p where down-egulated after acute exposure to cocaine. Moreover, they correlated these downregulated miRNAs to the upregulation of several networks associated with cocaine dependences comprehensive of PKC and JUN. Additionally, the study showed an interaction between miR-124-3p and TEAD1.

Another study, conducted by Chandrasekar et al. [53], observed significant downregulation of miR-124 in the caudate putamen region after chronic cocaine intake whilst let-7d was downregulated in the VTA, the CPU, HIP, PFC, and ROB regions. miR-181a was upregulated after chronic cocaine treatment in NAc, CPU, VTA, and piriform cortex. In a subsequent study, Chandrasekar et al. [54] evaluated the effect of miR-124 during the cocaine CPP acquisition period, suggesting its expression could be crucial for suppressing the reinstating properties of cocaine. Instead, once the CPP has been established, the overexpression of miR-124 did not affect the reinstatement of cocaine CPP. They found a correlation between miR-124, miR-181, and let-7d with the observed behavioral changes after cocaine CPP, showing that these miRNAs are modulated by cocaine CPP.

A study by Chen et al. [55] explored the changes in miRNA expression in the hippocampus of rats during the acquisition and extinction of cocaine-induced CPP. These miRNAs regulate the expression of target genes that act in many cellular processes most of which are involved in metabolic process and biological regulation. Some of these miRNAs were also implicated in brain disorders and drug abuse.

Interestingly, Chivero et al. [56] used EV loaded with miR-124 to restore miR-124 levels and thus function as a potential therapeutic approach for cocaine-mediated neuroinflammation. EV-loaded miR-124 inhibits cocaine-mediated microglial activation. In fact, cocaine determines the downregulation of miR-124 and, therefore, upregulates the expression of TLR4 and STAT3. On the other hand, Dash et al. [58] observed that miR-124 regulated PARP-1 by binding its 3′-UTR. Using a vector containing the 3′-UTR of PARP-1 in the presence or absence of miR-124, they found that knockdown of miR-124 resulted in a significant induction of PARP-1. The study suggested that PARP-1 3′-UTR activity is regulated by miR-124 expression in neuronal cells. Moreover, they found that miR-124 had a dominant regulatory effect compared to miR-125b. Conjointly downregulation of miR-124 is suggested in multiple neuroinflammatory diseases comprehensive of amyotrophic lateral sclerosis, PD, while its overexpression could mitigate the pathogenesis [75]. Guo et al. [66] observed that cocaine significantly decreased miR-124 expression in microglial cells by exposing BV-2 cells to cocaine for 24 h. Conversely, cocaine exposure increased levels of DNMT in rats’ primary microglia, and, in particular of DNMT1. Moreover, they discovered these alterations also in vivo. Then, they demonstrated that DNMT1 inhibitor 5-Azacytidine (5-AZA) reversed cocaine-mediated effects on miR-124 and DNMTs. These findings demonstrate a direct link between cocaine and the expression of miR-124 and DNMTs, therefore indicating that changes in the promoter of DNA methylation could be crucial in the cocaine-mediated downregulation of miR-124.

Dash et al. [57] evaluated the expression of miR-125b in a dopaminergic neuronal cell model and in the NAc of mice, finding it downregulated in both models. They also investigated PARP-1 expression finding that miR-125b negatively regulates PARP-1 acting on a post-transcriptional level. Moreover, it seems that miR-125b downregulation is dependent on DAT. Chronic cocaine exposure activated the miR-125b/PARP-1 axis, therefore, reducing its action on cellular damage repairs.

Doke et al. [60] evaluated miRNA variations in both HIV Tat 1 and cocaine-treated human primary astrocytes in order to understand if exposure to cocaine could dysregulate miRNA expression. They established that 8 miRNAs were significantly upregulated and 13 were downregulated when exposed to HIV Tat 1, while cocaine exposure altered the expression of 40 miRNAs, out of which 6 were downregulated while 8 were upregulated. Finally, they estimated how combined exposure altered miRNA expression, finding that they predominantly affected miR-155, miR-5580, miR-551a, miR-5090, miR-7155, miR-1255b2, miR-3133, and miR-6780a. miRNAs targeted gene sets associated with non-neoplastic disorder, unipolar disorder, and major depressive disorder and revealed they impacted most on pathways involved in metabolic pathways, fatty acid metabolism, and glycolysis/gluconeogenesis pathways. The correlation between cocaine exposure and HIV was also investigated by Napuri et al. [73] who evaluated the effect of cocaine on miR-155 and miR-20a. Both miRNAs are involved in the regulation of immune cell functions and are suppressed by HIV. They observed a significant downregulation of miR-155 and miR-20a after cocaine exposure. Then they focused on whether there was a synergic effect between cocaine and HIV on those miRNAs. Their findings suggested that cocaine exposure could enhance HIV-1 transcription and increase viral production in association with miR155 and miR-20a downregulation.

Sadakierska-Chudy et al. [78] assessed whether cocaine could induce long-lasting alterations of miRNA expression crucial for synaptic plasticity of the striatal region. They found miR-132 and miR-212 meaningfully upregulated after active cocaine intake. Their role is supposed to be crucial to synaptic plasticity and/or learning adaptation in the dorsal striatal region of the brain.

Schaefer et al. [79] found altered levels of 23 miRNAs. Moreover, they observed that Ago2 had a role in cocaine addiction due to ago2-dependent miRNAs and, specifically miRNAs with a potential role in neural plasticity and motivation to consume cocaine.

Tobon et al. [81] demonstrated that the D1 receptor is regulated in the caudate-putamen region of the brain but not in the nucleus accumbens and that D1 receptor levels increase within 30 min of cocaine administration in cocaine-sensitized mice. Within 5 min of cocaine administration, they found that levels of miR-142-3p and miR-382 were significantly decreased and that their downregulation in the caudate-putamen post-transcriptionally regulates D1 receptors.

Vannan et al. [80] studied the differential expression of miRNAs in the NAc shell and their correlation to cocaine-seeking behavior. They observed that 8 miRNAs were downregulated while 25 were upregulated in animals with low cocaine-seeking behavior. These miRNAs regulated the expression of several miRNAs targets: Zbtb20, Nfat5, Ago1, Qui, Nfib, Pde3a, Tcf4, Klf7, and Rorb.

Viola et al. [82] observed downregulation of miR-212 after 2 h cocaine-induced CPP test whilst Mecp2 is upregulated in the PFC of normally reared mice. This effect is blunted in maternally separated mice. Moreover, cocaine exposure determined a decreased expression of Bdnf.

Viola et al. [83] evaluated the association between cocaine abuse and miRNA expression showing upregulation of miR-124 and miR-181. These miRNAs regulate pathways involved in biological regulation, metabolic process, biogenesis, developmental process, response to stimulus, adhesion, signaling, and immune system.

### 3.3. miRNAs Expression in Opioids Abuse

Heroin is a morphine derivative, belonging to the opioids class, and it is one of the most powerfully addictive drugs of abuse in the world. Heroin addiction is a relapsing and chronic brain disorder that causes persistent alterations in synaptic plasticity [86].

Recent studies have indicated that miRNAs play a pivotal role in addiction by directly controlling synaptic remodeling, dendritic spine morphogenesis, drug dependence, and drug-rewarding properties in substance use disorders such as alcohol, cocaine, amphetamine and, of course, opioids [87] see Table 3.

Chronic morphine treatment has been shown to regulate the levels of miRNA-23b and let-7, which subsequently repress the µ-opioid receptor (MOR1) at a post-transcriptional level [98].

miRNAs offer a promising biomarker for integrating gene expression and regulation in the processes of substance use disorders.

Current research indicates that the NAc plays a central role in the regulation of behavioral functions associated with depression, anxiety, and addiction due to its involvement in the brain reward circuit. Approximately 95% of cells within the NAc are GABAergic neurons that possess dopamine D1-and D2-like receptors. NAc function is influenced by dopaminergic projections from the ventral tegmental area and numerous studies have indicated that the high level of MOR distribution in the NAc might be related to the behavioral properties of opioid addiction, including reward and withdrawal symptom [103].

The agonist of the µ-opioid receptor (OPRM1) elicits extracellular signal-regulated kinase (ERK) phosphorylation through different pathways: morphine utilizes the protein kinase C (PKC) pathway, whilst fentanyl operates in an arrestin2-dependent method.

Zheng and other authors, in their study, examined the effect of two pathways on miRNA expression. The first study [104] analyzed two pathways that led to distinct cellular localization of phosphorylated ERK and activated disparate sets of transcriptional factors. After treating the primary culture of rat hippocampal neurons and the mouse hippocampus with morphine or fentanyl for three days, they identified seven miRNAs regulated by one or both agonists. MiR-190 was downregulated by fentanyl but not by morphine. It was observed that the downregulation was attenuated by 1,4-diamino-2,3-dicyano-1,4-bis(methylthio)butadiene (U0126), a blocking agent of ERK phosphorylation. The observed effect was fentanyl-induced, but not morphine-induced. Downregulation of miR-190 was a consequence of selective ERK phosphorylation by the agonist. Additionally, the alterations in the expression of one of the miR-190 targets, NeuroD1, were found to correlate with changes in miR-190 expression. This suggests that the OPRM1 gene may regulate the NeuroD pathways by controlling the expression levels of miR-190. This aspect was expanded in a new study [105], in which it was shown that the cellular level of NeuroD1 is modulated differently: fentanyl increases NeuroD level by reducing the amount of miR-190, while morphine does not alter NeuroD level. After 3 days of treatment, morphine and fentanyl decreased the activity of Ca2/calmodulin-dependent protein kinase II (CaMKII), which phosphorylates and activates NeuroD. Zheng et al. [103] demonstrated that fentanyl decreases the miR-190 level by inhibiting the transcription of Talin2. Fentanyl-induced β-arrestin2-mediated ERK phosphorylation led to the phosphorylation of YinYang1 (YY1). In addition, YY1 phosphorylation impaired the association of YY1 with some regions on the Talin2 promoter, and this association was essential for YY1 to stimulate the transcription of the gene. Thus, fentanyl decreased the transcription of Talin2 and subsequently, the cellular level of miR-190 by inducing YY1 phosphorylation. In contrast, because morphine induces ERK phosphorylation via the protein kinase C pathway, morphine did not induce YY1 phosphorylation and had no effect on the transcription of Talin2 and the cellular content of miR-190.

Wang et al. [99] investigated the impact of opioids on the expression of anti-HIV miRNA in monocytes. Morphine-treated monocytes exhibited decreased levels of cellular anti-HIV miRNAs in comparison to untreated cells. Furthermore, morphine treatment of monocytes impeded the anti-HIV miRNA expression, induced by type I interferon (IFN). These findings align with the observation that morphine treatment of monocytes increases HIV replication. The action of morphine on the anti-HIV miRNAs and HIV could be countered by the opioid receptor antagonists (naltrexone or Cys2, Tyr3, Arg5, Pen7-amide). Additionally, the effect of morphine on miRNA expression in vitro was supported by the finding that anti-HIV miRNA levels were significantly reduced in heroin-dependent individuals in vivo (miRNA-28, 125b, 150, and 382) in peripheral blood mononuclear cells than the healthy subjects. These in vitro and in vivo findings indicate that opioid use impairs intracellular innate anti-HIV mechanisms in monocytes, contributing to cell susceptibility to HIV infection.

Another study [100] demonstrated that morphine treatment leads to differential miRNA and protein expression that have a potential impact on inflammation and oxidative stress processes that lead to the expansion of the HIV-1 viral reservoir in the CNS. Notably, hsa-miR-15b is likely to target FGF-2, resulting in the downregulation of FGF-2 secretion caused by morphine, and hsa-miR-15b was upregulated under similar treatment conditions. The induction of MCP-2 and IL-6 secretion following morphine treatment has been observed to elicit a pro-inflammatory response. Additionally, the unique aspects of morphine-induced metabolic changes in the CNS are represented by the induction of mitochondrial superoxide dismutase.

He et al. [106] examined the role of let-7 miRNAs in cellular and animal models of opioid tolerance. Inhibiting let-7 expression increased MOR protein expression in SH-SY5Y cells and the level of let-7 appeared to be inversely correlated with that of the receptor. On the other hand, let-7 expression was regulated by chronic treatment with morphine in SH-SY5Y cells and in mice that were tolerant to opioids. Moreover, knocking down let-7 partially attenuated the antinociceptive tolerance to morphine.

Morphine also decreases miR-133b expression, hence increasing the expression of its target, Pitx3, a transcription factor that activates tyrosine hydroxylase and dopamine transporter [93,107].

Yan et al. [95], revealed a functional role of miR-218 and its target, MeCP2, in the regulation of heroin-induced behavioral plasticity. MiR-218 was downregulated by chronic heroin use. MiR-218 is encoded by an intron of the Slit gene and inhibits the expression of Robo1 which play important roles in axonal growth, in motor neuron differentiation, and its loss of miR-218 cause systemic neuromuscular failure.

Jia et al. 100 [88] noted that repeated morphine exposure has been shown to induce neuronal plasticity in reward-related areas of the brain, particularly the dentate gyrus. miR-132, a CREB-induced and activation-dependent microRNA, has been suggested to be involved in neuronal plasticity by increasing neuronal dendritic branches and spinogenesis in rats treated with increasing doses of morphine injection for six consecutive days to develop morphine dependence. From their results, it is still unclear whether miR-132 is related to morphine dependence. However, they demonstrated that morphine treatment (24 h) promotes the differentiation of N2a cells stably expressing μ-opioid receptor by upregulating miR-132 expression. Moreover, inhibiting miR-132 3p (but not 5p) of the DG neurons can reverse the structural plasticity and disrupt the formation of morphine dependence in rats.

Chronic morphine intake induces a sustained increase in D1 receptor expression in glutamatergic terminals of medial prefrontal cortex (mPFC) projection neurons to the BLA but has no effect on D1 receptor expression in projection neurons from the hippocampus or thalamus to the BLA. This adaptation to chronic morphine is mediated by reduced expression of miR-105 in the mPFC, resulting in increased D1 receptor expression in glutamatergic terminals of projection neurons from the mPFC to the BLA. Projection neurons from the mPFC to the BLA are activated by abstinence-associated environmental cues morphine and overexpression of miR-105 in the mPFC leading to a reduction in D1 receptor induction [96].

Xie et al. [94], demonstrated that miR-592-3p, in the nucleus accumbens (NAc), can improve the incubation of morphine craving by targeting TMEFF1, and thus, it holds a therapeutic potential to inhibit opioid craving. Indeed, the miR-592-3p observed to be downregulated in the NAc core was linked to the incubation of morphine craving. Also, gain- and loss-of-function analyses revealed that the inhibition of miR-592-3p expression in the NAc core negatively regulated TMEFF1 expression, however, overexpression of miR-592-3p in the NAc nucleus resulted in decreased expression of TMEFF1, thereby reducing the incubation of morphine craving.

Michelhaugh et al. [101] described that also NEAT 1, NEAT 2, EMX2OS, MIAT, and MEG were upregulated in heroin abusers.

Different studies demonstrated how chronic morphine treatment increases let-7 and miR23b expression in a time- and dose-dependent manner and suppresses the association of MOR1 mRNA with polysomes. Expression of miR-24, miR-127, miR-186, and miR-222 was upregulated in the prefrontal cortex region of rats with escalated methamphetamine use. These findings suggested chronic exposure to addictive drugs may significantly alter brain miRNA profiles [90,106].

Liao et al. [92], examined the proinflammatory cytokine IL6/TNFα mRNA levels in isolated adult microglia from the brains of mice administered either morphine or saline, as an indicator of microglial activation. They demonstrate that miR-138 released in morphine-stimulated astrocyte EVs was taken up by the microglia, resulting in the activation of the TLR7/8 signaling pathway via the interaction of miR138 with TLR7.

Gu et al. [89] recruited 42 subjects who were addicted to heroin and from each subject was collected a total of 3 mL of venous blood. Microarray analysis identified 116 significantly altered miRNAs in heroin abusers; 3 miRNAs, including let-7b-5p, miR-206, and miR-486-5p, were verified to be significantly and steadily increased in heroin abusers, compared with normal controls.

### 3.4. miRNAs Expression in Methamphetamine Abuse

Methamphetamine (MA) is a synthetic drug of abuse with psychostimulant effects that is widely used because of its relatively low cost of manufacture. The chemical structure of methamphetamine differs from that of amphetamine by the addition of a single methyl group. This simple structural change gives the molecule enhanced neurostimulant potency, with effects that can last up to 8 h [108]. Like amphetamine, it increases motor activity, reduces appetite, and induces a general sense of well-being. The initial euphoria is followed by a marked state of agitation, which, in some individuals, may lead to violent behavior. In addition, like all substances of abuse, methamphetamine can easily lead to addiction, which, if prolonged, is typically associated with a spectrum of cognitive changes, particularly in learning and memory processes [109]. From a neurophysiological perspective, methamphetamine primarily affects the mesolimbic dopaminergic system, which is associated with motivation and reward pathways. Methamphetamine modulates the release of the neurotransmitter involved in these systems, dopamine, initially leading to increased release in certain brain areas such as the nucleus accumbens. However, during addiction, this sustained stimulation leads to chronic dopamine depletion over time, which appears to provide a neurophysiological explanation for the effects of dependence on this substance [110]. Indeed, several studies have shown that, methamphetamine dependence is associated with morphological and molecular changes in the brain, including changes in gene expression, as well as behavioral changes [111,112]. Recently, many scientists have focused on miRNAs to study their behavior and their effects on gene regulation in cases of methamphetamine abuse see Table 4.

Chand et al. [112] evaluated the role of an extracellular vesicle (EV) associated with microRNA-29a-3p in chronic methamphetamine use disorder. They demonstrated that EV-miR29a-3p elicited both pro-inflammatory and injurious responses via the TLR7 pathway. The authors also provided evidence of the therapeutic relevance of the anti-inflammatory drug Ibudilast, which was shown to rescue EV biogenesis, reduce the secretion of miR-29a, and rescue synaptodendritic injury. These data support the use of Ibudilast as a therapeutic intervention for methamphetamine use disorder. Finally, they showed that miR-29a is upregulated in plasma obtained from male participants identified with methamphetamine use disorder, indicating a possible role as a potential biomarker for therapeutic interventions.

Du and colleagues [113] screened 28 differentially expressed microRNAs (miRNAs) in controlled or escalated methamphetamine use in rats. They found that miR-186 increased and miR-195 and miR-329 decreased in the prefrontal cortex of rats that self-administered methamphetamine for 1 h per session (controlled group). In contrast, miR-127, miR-186, miR-222, and miR-24 increased and miR-329 decreased in rats that self-administered methamphetamine for 6 h per session (escalated group). The levels of miR-329 decreased similarly in both groups, while the increase in miR-186 was higher in the escalated group. Bioinformatic analysis indicated that the predicted targets of these verified miRNAs are involved in the processes of neuronal apoptosis and synaptic plasticity.

Sun et al. [114] conducted a case-control association study in 400 methamphetamine users with psychotic symptoms and 448 controls. They found that miR-181a expression is increased in methamphetamine use disorder with psychosis, suppressing the expression of GluA2 on hippocampal neurons.

Zhu [115] identified two novel miRNA candidates, novel-m002C and novel-m009C, which were upregulated in the nucleus accumbens of mice exposed to daily injections of methamphetamine. Additionally, scientists observed changes in two predicted gene targets, upregulation of Arc and downregulation of Pde4c, which have been implicated in addiction and brain plasticity after drug exposure.

Kim et al. [116] analyzed the expression of miR-137 in the circulating extracellular vesicles (EVs) of patients with methamphetamine abstinence. They observed a stable reduction in circulating miR-137 across more than 20 years of methamphetamine abstinence, with a diagnostic accuracy of up to 97.7%. However, the functional roles of circulating miR-137 in animals remain unexplored to date.

Zhao et al. [117] investigated the differential expression of plasma microRNAs (miRNAs) in 124 patients with methamphetamine use disorders. They found that the expression of miR-181a, miR-15b, miR-let-7e, and miR-let-7d in plasma was downregulated compared with normal controls. Additionally, they observed a negative correlation between the expression level of these miRNAs and drug use days in the past months, suggesting that more severe drug use was associated with more aberrant miRNA expression.

Demirel et al. [118] studied samples of the ventral tegmental area and nucleus accumbens in postmortem human brain tissue from methamphetamine users. Through quantitative reverse transcription PCR, they found overexpression of miRNA let-7b-3p, which could be used as a diagnostic and therapeutic marker in methamphetamine addiction.

Li [119] compared the expression of miRNAs in the serum exosomes of methamphetamine-dependent and ketamine-dependent rats and identified ten differentially expressed co-miRNAs in the two model groups. miR-128-3p, miR-133a-3p, miR-152-3p, and miR-181a-5p are overexpressed in methamphetamine-dependent rats. From KEGG pathway analysis, they found that these target genes are mainly located in SNARE interactions in vesicular transport, amphetamine addiction, cGMP-PKG signaling pathway, dopaminergic synapse, and GABAergic synapse.

Shang and his group [120] demonstrated that changes in Ppp3r1, Cdkn1c, Fmr1, and PPARGC1A expression were negatively correlated with miR-222-3p expression in methamphetamine-induced conditioned place preference. Their findings suggest that these miRNAs may be involved in nervous system development and in the regulation of methamphetamine-induced reward-related changes in the brain.

Yang and colleagues [108] developed a rat model of methamphetamine addiction and analyzed the expression profile alterations of miRNA in the nucleus accumbens of the methamphetamine-addicted rats. A total of 40 differential miRNAs were identified: 17 upregulated and 23 downregulated that responded to methamphetamine. They performed a bioinformatic analysis of the differential miRNAs that suggested that these miRNAs may contribute to methamphetamine-induced alterations in the nervous system through the Wnt signaling pathway, axon guidance, and lysosome pathways.

Sim et al. [121] determined the global miRNA expression profiling in Nucleus Accumbens of methamphetamine-addicted rats. MiR-496-3p, miR-194-5p, miR-200b-3p, and miR-181a-5p, were found to be significantly associated with addiction. Canonical pathway analysis revealed that a high number of methamphetamine addiction-related miRNAs play important roles in cell apoptosis, cellular defense, and neurogenesis pathways.

Moradali and colleagues [122] conducted a case-control study in Iran to investigate miR-127 and miR-132 expression in methamphetamine abusers. They observed the downregulation of miR-127 and upregulation of miR-132 in patients with methamphetamine abuse compared with the control group. Authors hypothesize a correlation with CREB, GnRH, and MAPK signaling pathways involved in synaptic transmission, angiogenesis, and inflammation.

The study conducted by Wang and his team [123] revealed that miR-181a may be indirectly responsible for methamphetamine addiction and downregulation of GABAAα1 through the regulation of ERAD. In fact, methamphetamine may activate the ERAD pathway through the downregulation of miR-181 and may induce the ubiquitination degradation of GABAAα1, which is involved in the formation of addiction.

Deng et al. [136] conducted a review to summarize the role of miRNAs in the context of methamphetamine use, particularly their involvement in the reward effect and neurotoxic sequelae. They reported that several miRNAs actively participate in neuroinflammation, including miR-155-5p [125] and miR-143 [124], which can inhibit specific proinflammatory gene pathways, such as Peli1, PUMA, and NLRP3. Furthermore, they highlighted that methamphetamine could induce changes in miRNA expression in specific brain sub-regions. One of the most extensively studied regions is the Nucleus Accumbens, as it is one of the primary areas involved in regulating reward pathways. In this region, they observed relevant changes in miRNA expression during methamphetamine sensitization in mice: downregulation of miR-124 [126] and miR-3068-5p [129], associated, respectively, with the overexpression of Dicer1 and Grin1 genes, both of which are involved in synaptic plasticity. Conversely, the upregulation of miR-29c [127], miR-204-3p [128], and miR-9-5p [121] can lead to differential expression of genes involved in brain physiology, structural plasticity, and behavioral sensitization, such as Dnmt3a, Dnmt3b, Meg3, Sema3A, Plxna4, and BDNF. Similar changes were observed in other brain regions, such as the Ventral Tegmental Area [130], Dorsolateral Striatum [131], and Hippocampus [132,134], where the downregulation of miR-145, miR-129, and miR-29, and the upregulation of miR-134, miR-31-3p, and miR-183-5p were associated with alterations in the expression of HDAC, LIMK1, RhoA, and NRG1 genes.

Zhang et al. [134] explored the relationship between miR-181a and the GRIA2 gene in chronic and acute methamphetamine users. They discovered that miR-181a significantly downregulates the expression of GRIA2, a gene involved in enhancing the effects of addictive drugs. The researchers observed that chronic methamphetamine use leads to a downregulation of miR-181a expression in human serum, resulting in an overexpression of GRIA2 proteins.

Kim and colleagues [135] investigated the bidirectional miR-137/SYNCRIP pathway during methamphetamine abstinence in animal models. They observed the upregulation of miR-137 in the brain (Dorsal Striatum) and a concurrent downregulation of the same miRNA in the blood. They demonstrated that the upregulation of miR-137 during abstinence triggers the inhibition of the SYNCRIP gene, which is associated with the cognitive symptoms of methamphetamine abstinence. However, the inhibition of this gene activates intracellular miR-137 expression and exosome release, leading to an overexpression of this miRNA in the blood.

## 4. Discussion

Addiction to drugs and alcohol represents a complex mental health disorder, and it has become a worldwide public health burden. This is why a thorough understanding of the molecular mechanisms that regulate such a disorder could be fundamental for its treatment.

A review of the current literature revealed that miRNAs play a key role in the modulation of genes, pathways, and neural circuits that control addiction and its various stages.

This work shows how miRNAs are deeply involved in the neuroadaptive responses induced by exposure to substances of abuse; in particular, studies on changes in expression of miRNAs determined by four substances were evaluated: alcohol, cocaine, methamphetamine, and opioids.

Hundreds of miRNAs were found to be dysregulated following the intake of such substances but, most importantly, it is interesting to notice how the expression of some miRNAs turned out to be phase-specific: some miRNAs have shown to be upregulated after an acute intake of substances—such as miR-124 or miR-132 that were upregulated after one hour of alcohol intake [26]—whereas other miRNAs’ expression appeared to be modified after chronic assumption of substances, or even in the withdrawal phase [40,44,46,69,88,116].

This work also highlighted how the expression of some miRNAs seems to be linked to the intake of specific substances Table 5, Table 6, Table 7 and Table 8: for example, the expression of miRNA 146a appeared to be upregulated only after alcohol intake [18], miR150 was altered after opioids’ consumption, overexpression of miR29c was only found after methamphetamine intake [137]. On the other hand, the altered expression of some other miRNAs (in particular, miR-206a-3p, miR-124a, miR-9, miR-132, miR-382, let-7d, let-7b, miR-153, miR-133b, miR-29b, miR-181, miR-186, and miR-152) was common subsequently to the intake of all four—or at least three substances.

Even if the processes by which miRNAs regulate and are regulated by such phenomenon remain difficult to understand, the literature showed that the most frequently mentioned targets were BDNF [4], CREB [67], and MeCP2 [68]. These pathways are known to play a crucial role in the development and maturation of the central nervous system, as also in neuronal differentiation and in formation of synapses. They are also involved in processes of plasticity and remodeling of the synapses.

Other molecules frequently targeted by miRNAs during addiction are Pitx3 [32], MAPK signaling pathway [2], and other genes involved in the processes of cell repair, neuroinflammation, apoptosis, and even in metabolic processes such as glycolysis and fatty acid metabolism [60,100].

Such pleiotropy could partly explain how altered miRNAs’ expression induced by intake of substances of abuse can determine both short-term and long-term effects on nerve functions of attention, learning, memory, behavior, and motivation. Not surprisingly, the brain regions in which miRNAs were most frequently isolated were the Nucleus Accumbens (Nac), the dorsolateral striatum (DLS), and the Ventral Tegmental Area (VTA)—some of the primary areas involved in regulating reward pathways. They were also found in the prefrontal cortex and in the hippocampus.

Interestingly, miRNAs altered in addiction were also found dysregulated in psychiatric and neurodegenerative disorders, such as schizophrenia, major depression, Alzheimer’s, and Parkinson’s disease, as such conditions determine the involvement of the same circuits and processes of addiction [22,25].

Most of the studies evaluated were conducted on animal models but, thanks to the high stability of miRNAs and the fact that they are highly conserved in human species, it has been demonstrated that they can be considered as potential biomarkers for the diagnosis of addiction. This is made possible by the fact that some miRNAs altered in addiction, could be easily extracted and measured in body fluids such as blood (as miR-181 and 221, or miR-124) [31,82], and in saliva [48].

Finally, studies conducted by Bahi et al. [21] and Bekdash et al. [38] also demonstrated that, by modulating the expression of certain miRNAs, it is possible to reduce the consumption of the substances, or even turn down the effects of withdrawal, thus suggesting a role for these little molecules as potential therapeutic tools [2].

## 5. Conclusions

Drugs and alcohol addiction represent a complex mental health disorder, and it has also become a worldwide public health burden. However, its diagnosis is still based on clinical criteria. With this work, we aimed to collect and unify the current knowledge on the role of miRNAs in addiction, in order to elucidate the biological processes that regulate this disorder. In recent years, miRNAs have increasingly attracted the attention of scientists and researchers due to their cardinal role in regulating biological processes, including neurogenesis, synaptic plasticity, and even neuroinflammation and apoptosis. Our study has shown that the major targets of miRNAs during addiction appear to be those involved in the development and maturation of the central nervous system, but also those related to cell cycle regulation and metabolism. It seems that the areas most frequently involved in such processes, are those of the reward circuit, and also those involved in regulating functions such as behavior, motivation, attention, and memory. In many cases, it was observed that certain miRNAs can be used as diagnostic markers for subsequent clinical outcomes. Moreover, it was observed how their pharmacological regulation could lead to tangible therapeutic benefits. However, this work shows that our biological and molecular understanding of addiction is not yet fully known. Therefore, further studies on this topic will be needed with the aim of the development of new diagnostic and effective therapeutic methods.

## Figures and Tables

**Figure 1 ijms-24-17122-f001:**
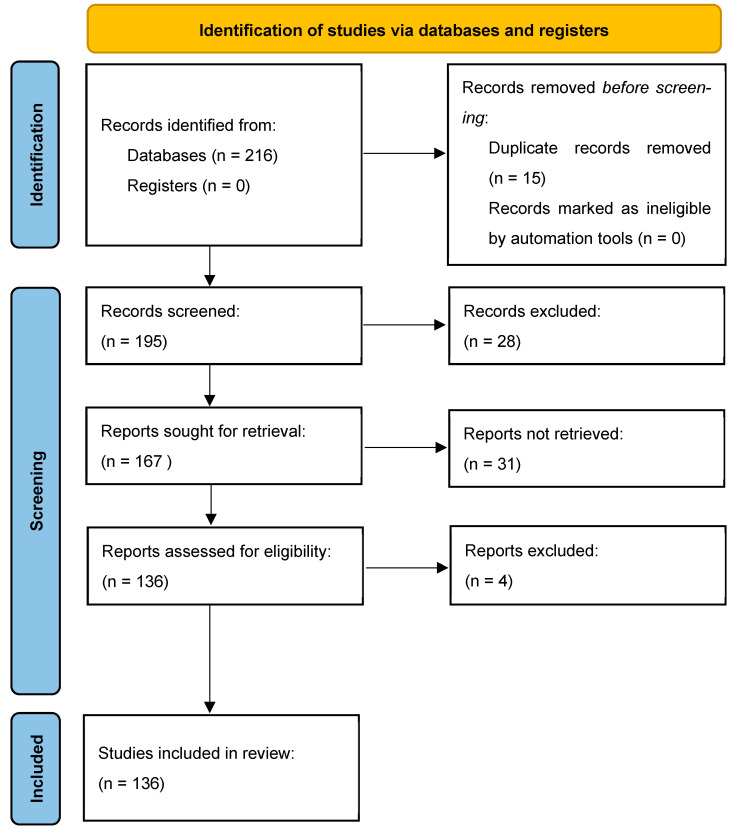
The selection of appropriate scientific papers was performed; articles met the inclusion criteria and were included.

**Table 1 ijms-24-17122-t001:** miRNAs expression in Alcohol abuse.

Reference	miRNA	Target	Expression (↓↑*)	Effect in Alcohol Abuse
Ehinger et al. (2020) [19]	miR30a-5p miR206-3p miR191-5p miR195-5p	BDNF/TrkB pathway	↑ miR30a-5p, miR206-3p, miR191-5p, and miR195-5p ↓ BDNF	miR30a-5p, miR206, miR191, and miR195 are upregulated after 8 weeks of heavy alcohol consumption. Upregulation is accompained by a reduction of BDNF levels.
Tapocik et al. (2013) [20]	miR206	BDNF	↑ miR206 ↓ BDNF	miR206 overexpression suppresses BDNF. Inhibition of miR206 determines inversion of BDNF repression.
Bahi and Dreyer (2013) [21]	miR124a	BDNF	↓ miR124a ↑ BDNF	miR124a is downregulated after 15 days of voluntary ethanol drinking, whereas BDNF levels are increased. Downregulation of miR124a determines reduction of alcohol intake.
Darcq et al. (2015) [22]	miR30a-5p miR195-5p	BDNF	↑ miR30a-5p and miR195-5p ↓ BDNF	miR30a-5p and miR195-5p are upregulated during binge drinking and abstinence. Upregulation of miR-30 a-5p and miR195-5p downregulates BDNF.
Pietrzykowski (2012) [23] Gu et al. (2018) [24] Li et al. (2011) [25]	miR9	Alpha subunit BK channel; HDA5; DRD2	↑ miR9 ↓ BK channel ↓ HDA5 ↓ DRD2	miR9 is upregulated after acute exposure to alcohol. However, downregulation of DRD2 occurs after a chronic assumption of the substance.
Mizuo et al. (2016) [26]	miR124a miR132	HDAC4	↑ miR124a and miR132 ↓ HDAC4	miR-124 and miR-132 were upregulated 1 h after ethanol administration and remained elevated for a full 12 h.
Lewohl et al. (2011) [18]	miR203/miR586/ miR146a miR519b-3p/miR665	PLP1 CNP	↑ miR203, miR586 and miR146a ↓ PLP1 ↑ miR519b-3p, and miR665 ↓ CNP	miRNAs and their targets’ expression usually is inversely proportional. This means that when a miRNA is upregulated, its targets are downregulated and vice versa.
Li et al. (2013) [27]	miR-382	DR1	↓ miR-382 ↑ DR1	miR-382 is downregulated after repeated exposure to alcohol, while DR1 is upregulated. Overexpression of miR-382 determines downregulation of DR1.
Bahi et al. (2013) [21]	Let-7d	DR3	-	Overexpression of let-7d induces downregulation of DR3.
Most et al. (2016) [28]	miR-411	GluR2	↓ miR-411 ↑ GluR2	Downregulation of miR-411 reduces GluR2 expression and alcohol assumption.
Coleman et al. (2017) [29]	Let-7b	TLR7	↑ let-7b ↑ TLR7 ↑ NF-kβ	Increased release of let-7b determines an hyperactivation of the TLR7-NF-kβ pathway.
Balaraman et al. (2012) [30]	miR9; miR21; miR153; miR335; miR140-3p	-	-	These miRNAs, regulated by alcohol intake, can cross the placenta and exert their effects on the fetus.
Alhaddad et al. (2020) [31]	miR-124-3p	GRα	↑ miR-124-3p ↓ GRα	Upregulation of miR-124-3p following long alcohol exposure determines reduction in GRα levels.
Lim et al. (2021) [32]	miR133b	Pitx3	↑ miR-133b ↓ Pitx3	Upregulation of miR133b, reduces the expression of Pitx3.
Tang et al. (2008) [33]	miR-212	ZO-1	↑ miR-212 ↓ ZO-1	Upregulated by alcohol, determines the reduction of ZO-1.
Tseng et al. (2019) [34]	miR140	-	-	Upregulated in microvesicles released by fetal neural stem cells.
Darcq et al. (2015) [22]	miR1	BDNF	-	miR1 is upregulated by alcohol.
Sathyan et al. (2007) [35]	miR21 miR335	GABA_A_	↓ miR21 ↓ miR335	miR21 and miR335 were suppressed following alcohol transfection. miR21 prevents apoptosis and its suppression determines cell death. miR335 is an apoptosis permissive factor.
Qi et al. (2014) [36]	miR29b	SP1/RAX/PKR	↓ miR29b ↑ SP1	miR29b is suppressed after ethanol exposure, whereas SP1 is upregulated. Administration of miR29b mimic exerts a protective effect against apoptosis.
Asquith et al. (2014) [37]	miR181 miR221	STAT3/ARNT	↑ miR181 and miR221 ↓ STAT3 and ARNT	miR181 and miR221 were upregulated after self-administration of ethanol. Such featuring is accompanied by the downregulation of STAT3 and ARNT.
Bekdash et al. (2015) [38]	miR186 miR24 miR375 miR155	Gabra4	↑ miR186, miR24, miR375, and miR155 ↓ Gabra4	miR186, miR24, miR375, and miR155 were upregulated in neurons that were withdrawn from chronic alcohol exposure for 8 h. Upregulation of such miRNAs determined reduction of Gabra4 levels.
Choi et al. (2020) [39]	miR125a-3p	Prdm5 Suv39h1 Ptprz1 Mapk9	↑ miR125a-3p ↓ Prdm5, Suv39h1, Ptprz1, and Mapk9	Chronic alcohol exposure determined induction of miR-125a-3p and downregulation of its targets in male rats.
Choi et al. (2020) [39]	Let-7a-5p	Lin28a Apbb3 Nras Acvr1c	↑ Let-7a-5p ↓ Lin28a, Apbb3, Nras, and Acvr1c	Chronic alcohol exposure determined induction of Let-7a-5p and downregulation of its targets in male rats.
Choi et al. (2020) [39]	miR-881-3p	Naa50 Clock Cbfb	↓ miR-881-3p ↑ Naa50, Clock, and Cbfb	Chronic alcohol exposure determined inhibition of miR-881-3p and upregulation of its targets in female rats.
Choi et al. (2020) [39]	miR-504	Ar1h1 Ube2g1 Gng7	↓ miR-504 ↑ Ar1h1, Ube2g1, and Gng7	Chronic alcohol exposure determined inhibition of miR-504and upregulation of its targets in female rats.
Guo et al. (2012) [40]	miR-152 miR-199a-3p miR-685	MeCP2	↑ miR-152, miR-199a-3p, and miR-685 ↓ MeCP2	Chronic intermittent ethanol exposure determines upregulation of miR-152, -199a-3p, and -685.
Santos-Bezerra et al. (2021) [41]	miR-34a miR-34c	-	↑ miR-34a and miR-34c	miR-34a and miR-34c were found upregulated in subjects with AUD.
Manzardo et al. (2013) [42]	miR-3065-5p, miR-299-5p, miR-767-5p, miR-375, miR-29b, miR-377, miR-399, miR-572, and miR-3162	THBS2, CHN2, NDE1, ELMO1, TMCC3, SEMA4D1, UGT8, CLCA4, TMTC2	↑ miR-3065-5p, ↑ miR-299-5p, ↑ miR-767-5p, ↑ miR-375, ↑ miR-29b, ↑ miR-377, ↑ miR-399 ↓ miR-572, ↓ miR-3162	Twelve miRNAs were found upregulated, whereas two miRNAs were found downregulated. Targeted genes were involved in different functions as cellular adhesion, tissue and oligodecrocyte differentiation, neuronal migration, myelination.

* Abbreviation: ↓ downespression ↑ upespression.

**Table 2 ijms-24-17122-t002:** miRNAs Expression in Cocaine abuse.

Reference	miRNA	Target	Expression (↓↑*)	Effect in Cocaine Abuse
Barreto-Valer et al. (2012) [50]	miR-133b	Pitx3	↓ miR-133b ↑ pitx3	miR-133b may be involved in skeletal muscle development in embryos. miR-133 modulates the expression of Pitx3 and therefore of dopamine receptors DRD2, DAT and TH.
Bastle et al. (2018) [51]	miR-495	BDNF, Camk2a, and Arc	↓ miR-495 ↓ Bdnf ↓ Camk2a ↓ Arc	miR-495 overexpression increases motivation for cocaine by targeting BDNF, Camk2a, and Arc in the NAc.
Cabana-Dominguez et al. (2018) [52]	miR-9, miR-153 and miR-124	PKC, JUN, and TEAD1	↓ miR-9 ↓ miR-153 ↓ miR-124	miR-9, miR-153, and miR-124 are associated to cocaine-dependence networks.
Chandrasekar et al. (2009) [53]	miR-124, let-7d, and miR-181a	DA D3R	↓ miR-124 ↓ let-7d ↑ miR-181a ↑ DA D3R	miR-124, let-7d, and miR-181a modulates the expression of DA D3D implicated in synaptic plasticity.
Chandrasekar et al. (2011) [54]	miR-124, let-7d, and miR-181a	CREB, FosB, uPA, DAT, MOR1, Per2, 7MYT1/NZF-2b, Drd2, Drd3, BDNF, GRIA2	↓ miR-124 ↓ let-7d ↓ Per2 ↓ Drd2 ↓ Drd3 ↓ 7MYT1/NZF-2b ↓ MOR1 ↓ BDNF ↓ GRIA2 ↑ miR-181a, DAT, uPA	miR-181a enhanced cocaine-cue-induced stimuli. miR-124 could be crucial in reinstating the properties of cocaine after CPP is established. Finally, cocaine proved to modulate the expression of plasticity genes at the protein level.
Chen et al. (2013) [55]	miR-133b, miR-152, miR-144, miR-451, miR-770, miR-194, miR-22, miR-30a, miR-347, and miR-874	-	↓ miR-144 ↓ miR-451 ↓ miR-770	miRNAs expression modulated by cocaine is involved in the modulation of multiple target genes that regulate metabolic process and biological regulation.
Chivero et al. (2020) [56]	miR-124	TLR4, MYD88, Nf-kB p65, and STAT3	↓ miR-124 ↑ TLR4 ↑ MYD88 ↑ Nf-kB p65 ↑ STAT3	miR-124 attenuated cocaine-mediated upregulation of target genes therefore inhibiting microglial activation.
Dash et al. (2017) [57]	miR-125b	PARP-1	↓ miR-125b ↑ PARP-1	Chronic cocaine exposure activated the miR-125b/PARP-1 axis, acting negatively on cellular damage repair.
Dash et al. (2020) [58]	miR-124	PARP-1	↓ miR-124 ↑ PARP-1	Cocaine persistently reduces miR-124 levels after acute and chronic exposure. Therefore, reduced levels of miR-124 are present in animals that display cocaine-induced behavioral effects.
Doke et al. (2021) [59]	miR-135-2, miR428e, miR5580, and miR6799	VAMP2, PSD93, MEIS1, and NFIB	↑ miR-135-2 ↑ miR-6889 ↑ miR428e ↑ VAMP2. ↓ miR5580 ↓ miR-597 ↓ miR6799 ↓ MEIS1 ↓ NFIB ↓ PSD93	miRNA altered by cocaine and HIV Tat 1 are also associated with non-neoplastic disorder, unipolar disorder and major depressive disorder. They also revealed their involvement in metabolic pathways, fatty acid metabolism, and Glycolysis/gluconeogenesis pathways.
Doke et al. (2022) [60]	miR2355 and miR-4726-5p	NDUFA9, LIPG, KYNU, and HKDC1	↓ miR2355 ↓ miR-4726-5p ↑ has-miR-4726-5p ↓ NDUFA9	When LINC01133 is knockdown in astrocytes, miR-4276 is upregulated while NDUFA9 is upregulated. They hypothesize that the LINC01133-miR-4276-NDUFA9 axis could be involved in cocaine and HIV-1 induced astrocytes dysfunction and neurodegeneration.
Domingo-Rodriguez et al. (2021) [61]	miR-1249-3p and miR-34b-5p	MAPK signaling pathway	↓ miR-1249-3p ↓ miR-34b-5p	miR-34b-5p and miR-1249-3p are associated with high motivated and high motor disinhibited behavior.
Dykxhoorn et al. (2023) [62]	miR-423-5p and miR-30c	Cacna2d2, Me-1	↑ miR-30c ↑ Cacna2d2. ↓ miR-423-5p ↓ Me-1	miR-423-5p determines cocaine-induced smooth muscle cell contraction.
Forget et al. (2022) [63]	miR-1, miR-16, miR-29b, miR-31, miR-32, miR-124, miR-125b, miR-128b, miR-132, miR-181a, miR-128b, and let-7d	Fosb, BDNF, and Npas4	↑ miR-1 ↑ miR-16 ↑ miR-29b ↑ miR-31 ↑ miR-32 ↑ miR-124 ↑ miR-125b ↑ miR-128b ↑ miR-132 miR-181a ↑ miR-128b ↑ let-7d	miR-1 overexpression decreases cocaine-induced reinstatement of cocaine-seeking behavior.
Gawlinski et al. (2022) [64]	miR-374 and miR-544	Wnt5a, Wnt7b, and Ctnnb1	↓ miR-374 ↓ miR-544 ↓ Wnt5b ↑ Ctnnb1	Cocaine downregulated miR-374 and miR-544. In the paper, possible miRNA targets are not discussed, but they hypothesized a role of cocaine in the Wnt signaling pathway after chronic exposure.
Giannotti et al. (2014) [65]	Let-7d, miR-124, and miR-132	BDNF	↑ BDNF ↓ Let-7d ↓ miR-124 ↓ miR-132	miR-124, miR-132, and let-7d are downregulated after cocaine exposure whilst BNDF is overexpressed. They hypothesized that this overexpression of BDNF could have an impact on brain homeostasis after a withdrawal period.
Guo et al. (2016) [66]	miR-124	DNMT	↓ miR-124 ↑ DNMT	miR-124 downregulation after 24 h cocaine exposure may be linked to epigenetic changes cocaine-mediated on promoters of DNA methylation such as DNMTs.
Hollander et al. (2010) [67]	miR-212	CREB	↑ miR-212 and CREB	miR-212 may represent an anti-addiction counter-adaptive response in brain circuitries.
Im et al. (2010) [68]	miR-212	MeCP2 and BDNF	↑ miR-212 and BDNF	miR-212-MeCP2 interactions control cocaine upregulation of BDNF. BDNF facilitates compulsive cocaine-taking behavior.
Kumaresan et al. (2023) [69]	rno-miR-338-3p, rno-miR-935, rno-miR-382-3p, and rno-miR-770-5p	KEGG pathways	↑ rno-miR-338-3p, rno-miR-935, rno-miR-770-5p, and KEGG pathways ↓ rno-miR-382-3p	Rno-miR-101a-3p and rno-miR-137-3p could influence cocaine use by changing motivation to seek cocaine.
Li et al. (2020) [70]	miR-134	CREB, BDNF, SNAP23, STXBP5, and TICAM2	↑ miR-134 and ↓ CREB, BDNF, SNAP23, STXBP5, and TICAM2	miR-134 enhanced anxiety-like and depression-like behaviors in mice.
Lòpez-Bellido et al. (2012) [71]	let-7d	Zfdor1 and zfdor2	↑ let-7d ↑ zfdor1 ↑ zfdor2	Let-7d increased after cocaine exposure. Cocaine affects the levels of opioid receptors in the developmental stage in the neural plate, cephalic structures of the embryonic axis, posterior trunk, and tail bud regions of zebrafish embryos.
Mantri et al. (2012) [72]	miR-125b	-	↓ miR-125b	Cocaine enhanced the replication of HIV-1 in primary CD4+ T cells by downregulating miR-125b.
Napuri et al. (2013) [73]	miR-155 and miR-20a	PU.1	↓ miR-155 ↓ miR-20a ↑ PU.1	Cocaine exposure synergically downregulated miR-155 and miR-20a. The study suggested that cocaine exposure may enhance HIV-a transcription and viral production.
Nudelman et al. (2009) [74]	miR-132	CREB	↑ miR-132	miR-132 increased with age, therefore confirming its experience-dependent development.
Periyasamy et al. (2018) [75]	miR-124	KLF4 and TLR4 signaling	↓ miR-124 ↑ KLF4	miR-124 downregulated after cocaine expression determined upregulation of KLF4. It is suggested that cocaine action on miR-124 may be correlated with neuroinflammation. Moreover, miR-124 could mitigate the pathogenesis of neuroinflammatory diseases.
Quinn et al. (2015) [76]	miR-431 and miR-212	Arc	↑ miR-431	Reduced expression of miR-212 in addiction-vulnerable mice.
Quinn et al. (2018) [77]	miR-101b, miR-137, and miR-132	Drd1	↓ miR-101b ↓ miR-137 ↓ miR-132 ↓ Drd1	miR-212, miR-101-b, miR-137, and miR-132 increased in addiction-prone rats.
Sadakierska-Chudy et al. 2017 [78]	miR-212 and miR-132	Ago2, Pum2, REST	↑ miR-212 ↑ miR-132 ↑ Ago2 ↑ REST ↑ Pum2	miR-132 and miR-212 are upregulated after active cocaine intake. Their role is supposed to be crucial to synaptic plasticity and/or learning adaptation in the dorsal striatal region of the brain.
Schaefer et al. (2010) [79]	miR-467, miR-544, miR-154, miR-467b, miR-500, miR-186, miR-337-3p, miR-138, miR-369-3p, miR-136, miR-7a, miR-137, miR-324-5p, miR-665, miR-380-3p, miR-376a, miR-130a, miR-301b, miR-148b, miR-488, miR-384-5p, miR-376c, and miR-181a	Ago2, Drd2	↓ miR-467 ↓ miR-544 ↓ miR-154 ↓ miR-467b ↓ miR-500 ↓ miR-186 ↓ miR-337-3p ↓ miR-138 ↓ miR-369-3p ↓ miR-136 ↓ miR-7a ↓ miR-137 ↓ miR-324-5p ↓ miR-665 ↓ miR-380-3p ↓ miR-376a ↓ miR-130a ↓ miR-301b ↓ miR-148b ↓ miR-488 ↓ miR-384-5p ↓ miR-376c ↓ miR-181a ↓ Ago2	Downregulation of these 23 miRNAs is implied in cocaine addiction due to their Ago2 dependency. Their role could potentially be important in neuronal cells plasticity and motivation to consume cocaine.
Vannan et al. (2021) [80]	miR-463-3p, miR-346-5p, miR-483-3p, miR-3557-5p, miR-193a-3p, miR-133a-3p, miR-142-5p, and miR-3573-5p miR-29a-3p, miR-16-5p, miR-93-5p, miR-495-3p, miR-376c-3p, miR-410-3p, miR-329-3p, let-7a-5p, miR-652-3p, miR-377-3p, miR-107-3p, miR-138-5p, miR-487b-3p, miR-344v-1-3p, miR-128-3p, miR-137-3p, miR-323-3p, miR-337-3p, miR-125b-3p, miR-409-5p, miR-218a-5p, miR-212-3p, miR-130b-3p, miR-221-3p, and miR-132-3p	Nfib, Zbtb20, Nfat5, Ago1, Qki, Pde3a, Rorb, Tcf4, and Klf7	↓ miR-29a-3p, miR-16-5p, miR-93-5p, miR-495-3p, miR-376c-3p, miR-410-3p, miR-329-3p, let-7a-5p, miR-652-3p, miR-377-3p, miR-107-3p, miR-138-5p, miR-487b-3p, miR-344v-1-3p, miR-128-3p, miR-137-3p, miR-323-3p, miR-337-3p, miR-125b-3p, miR-409-5p, miR-218a-5p, miR-212-3p, miR-130b-3p, miR-221-3p, and miR-132-3p ↑ miR-463-3p, miR-346-5p, miR-483-3p, miR-3557-5p, miR-193a-3p, miR-133a-3p, miR-142-5p, and miR-3573-5p	Low cocaine-seeking mice showed downregulation of 8 miRNAs and upregulation of 25 miRNAs. These miRNAs are involved in the regulation of multiple pathways which regulate inflammatory response and oxidative stress.
Tobon et al. (2015) [81]	MiR-142-3p and miR-382	D1 receptor	↓ miR-142-3p ↓ miR-382 ↑ D1 receptor	D1 receptor levels increase within 30 min of cocaine administration in cocaine-sensitized mice. Downregulation of miR-142-3p and miR-382 act on a post-transcriptional level to increase expression in D1 receptor in the caudate-putamen region of the brain.
Viola et al. (2016) [82]	miR-212	BDNF and MeCP2	↑ miR-212, MeCP2, and BDNF	Upregulation of miR-212 2 h after cocaine induced CPP test while MeCP2 levels decreased in the PFC.
Viola et al. (2019) [83]	miR-124, miR-181, and miR-212	-	↑ miR-124 and miR-181	Upregulation of miR-124 and miR-181 after cocaine exposure in cocaine users. miR-124 and miR-181 regulated pathways involved in biological regulation and metabolic process.
Xu et al. (2013) [84]	miR-212	CREB	↑ miR-212	Upregulation of miR-212 after cocaine use increased cocaine activity on CREB. Potential role on metabolic processes.
Zhu et al. (2018) [85]	miR-30c-5p	Me1	↑ miR-30c-5p ↓ Me1	Upregulation of miR-30c-5p after cocaine exposure in mice aorta while Me1 was downregulated therefore determining ROS elevation.

* Abbreviation: ↓ downespression ↑ upespression.

**Table 3 ijms-24-17122-t003:** miRNAs Expression in Opioids abuse.

Reference	Target	Expression (↓↑*)	Effect in Heroine Abuse
Jia et al. (2022) [88]	MOR	↑ miR-32 ↑ MOR	miR-132 could mediate neuronal differentiation induced by 24 h morphine exposure in vitro.
Gu et al. (2020) [89]	TLR7 BDNF	↑ let-7b-5p-3p ↑ miR-486-5p ↑ miR-206 ↓ BDNF ↑ TLR7	let-7b-5p is a brain-specific miRNA involved in neuronal development and differentiation; when it is upregulated, it can attenuate opioid antinociceptive tolerance. miR-206 was recognized as a cerebellum-enriched miRNA and was found to be dysregulated in ketamine- treated rats by directly targeting brain-derived neurotrophic factor (BDNF). miR-486-5p may serve as novel biomarkers for monitoring nervous system impairments and function as surrogate markers to identify underlying neurophysiological processes of drug addiction.
Liu et al. (2021) [90]	MOR	↑ let-7b-5p ↑ mrR-320a ↓ MOR	Let-7b-5p plays a significant role in neurogenesis and synapse formation. MiRNA-320a is involved in depression, schizophrenia, cerebral ischemia, and neurodegenerative diseases.
Wenjin et al. (2020) [91]	BDNF CREB D3R P53 pathway MOTR TGF-β pathway	↑ hsa-miR-181a	The aberrant expression of miR-181a is involved in the neuronal apoptosis, synapse plasticity and neurogenesis, and immune function.
Liao et al. (2020) [92]	TLR7	↑ m-RNA-138 ↑ TLR7	The upregulation of miR138 in morphine-stimulated astrocyte EVs can be shuttled into microglia, resulting in activation of these latter cells. In agreement with the in vitro results, our in vivo data also demonstrated increased activation of microglia that was accompanied with morphological alterations in the thalamus of morphine-administered mice.
Hsu et al. (2019) [93]	Pitx3 BDNF Mu-opioid receptor	↓ miR-133 ↑ Pitx3	miR-133b levels may be lower in OUD patients on MMT.
Xie et al. (2022) [94]	TMEFF1	↑ miR-592-3p ↓ TMEFF1	The expression of TMEFF1 at protein levels was significantly upregulated in the cue-induced incubation of morphine craving in the NAcC. The downregulation of TMEFF1 could significantly decrease the morphine incubation, and its overexpression significantly increased morphine incubation, suggesting a functional role of TMEFF1 in cue-induced incubation of morphine craving. The overexpression of miR-592-3p decreased TMEFF1 protein levels in the incubation of morphine craving and its downregulation led to a significant increase of TMEFF1 protein levels, which indicated that miR-592-3p has a functional part in the regulation of TMEFF1 expression during the incubation of morphine craving.
Yan et al. (2017) [95]	MeCP2	↓ miR-218 ↓ MeCP2	miR-218 was downregulated by chronic heroin use in NAc. MiR-218 inhibits heroin-induced CPP by directly targeting MeCP2.
Zhao et al. (2017) [96]	D1R	↑ miR-105	Chronic morphine inhibits the expression of miR-105 in the mPFC, which results in enhanced D1 receptor expression in glutamatergic terminals of projection neurons from the mPFC to the BLA.
Kim et al. (2018) [97]	Wnt, MAPK, TGF-β, KEGG and neurotrophin signaling pathways	↑ mmu-miR-695, mmu-miR-32-5p, mmu-miR-202-5p, mmu-miR-6899-5p, mmu-miR-7231-5p, mmu-miR-201-5p, mmu-miR-6418-5p, mmu-miR-144-5p, mmu-miR-7042-5p, mmu-miR-496b, mmu-miR-6998-5p, mmu-miR-6416-3p, mu-miR-669d-2-3p, mmu-miR-5619-5p, mmu-miR-669d-3p, mmu-miR-6393, mmu-miR-8117, mmu-miR-3569-5p, mmu-miR-669j, mmu-miR-7670-5p, mmu-miR-155-3p, mmu-miR-7578, mmu-miR-8113, mmu-miR-7217-3p, mmu-miR-669m-3p, mmu-miR-327, mmu-miR-546, mmu-miR-7043-5p, mmu-miR-7118-5p, mmu-miR-6990-3p, mmu-miR-7038-3p, mmu-miR-7007-5p, mmu-miR-451a ↓ mmu-miR-3102-3p.2-3p, mmu-miR-1298-5p, mmu-miR-3068-5p, mmu-miR-181a-1-3p, mmu-miR-598-5p, mmu-miR-20b-5p, mmu-miR-344g-5p, mmu-miR-668-5p, mmu-miR-5122, mmu-miR-410-5p, mmu-miR-872-5p, mmu-miR-1946b, mmu-miR-669a-5p, mmu-miR-669p-5p, mmu-miR-28a-3p, mmu-miR-222-5p, mmu-miR-322-3p, mmu-miR-1896, mmu-miR-199a-3p, mmu-miR-199b-3p, mmu-miR-24-1-5p, mmu-miR-3110-3p, mmu-miR-344d-3-5p, mmu-miR-28c, mmu-miR-135a-1-3p, mmu-miR-3069-3p, mmu-miR-1982-3p, mmu-miR-1981-3p	15 miRNAs and 45 of their putative targets appear to regulate the Wnt, MAPK, TGF-β, and neurotrophin signaling pathways, which are involved in diverse neuronal functions, such as neurogenesis, synaptic plasticity, and neuroinflammation. Some of the miRNAs had multiple targets in each signaling pathway. It is also involved in the mouse model of morphine addiction the KEGG pathways.
Wu et al. (2009) [98]	MOR	↑ miR-23b ↓ MOR	Morphine induced a dose-dependent increase of miRNA23b and leads to a decrease in the polysome association of MOR1 mRNA. This effect was observed only in native MOR1 mRNA.
Wang et al. (2011) [99]	IFN-β IFN-α	↓ miR-28 ↓ miR-125b ↓ miR-150 ↓ miR-382	Heroin-dependent subjects had significantly lower levels of anti-HIV miRNAs (miRNA- 28, 125b, 150, and 382) in peripheral blood mononuclear cells than the healthy subjects.
Dave and Khalili (2010) [100]	FGF-2 IL-6 MCP-2	↑ miR-15-b ↓ miR-18-b	miR-15b and hsa-miR-181b have several predicted gene targets involved in inflammation and T-cell activation pathways. In this context, we observed induction of MCP-2 and IL-6 by morphine.
Michelhaugh et al. (2011) [101]		↑ Every lncRNA	These lncRNA are upregulated in heroine abusers.
Wu et al. (2008) [102]	MOR1	↑ miR-23b ↓ MOR1	Morphine induced a dose-dependent increase of miRNA23b and leads to a decrease in the polysome association of MOR1 mRNA. This effect was observed only in native MOR1 mRNA.
Zheng et al. (2010) [103,104,105]	Talin 2 YY1 β-arrestin2 ERK	↓ miR-190	Fentanyl downregulated miR-190 resulting in induction of NeuroD.
He et al. (2010) [106]	MOR	↓ MOR	Morphine significantly upregulated let-7 expression. Chronic morphine did not change the overall MOR transcript.
Sanchez-Simon et al. (2010) [107]	ERK1/2 TH DAT	↓ miR-133b	Morphine downregulated the miR-133b and it induces Pitx3. This caused an alteration in dopaminergic differentiation.

* Abbreviation: ↓ downespression ↑ upespression.

**Table 4 ijms-24-17122-t004:** miRNAs Expression in Methamphetamine abuse.

Reference	miRNA	Target	Expression (↓↑)	Effect in Methamphetamine Abuse
Chand et al. (2021) [112]	miR-29a-3p	TLR7	↑ miR-29a-3p ↑ TLR7	miR-29a-3p significantly increased in brain derived extracellular vesicles in chronic methamphetamine exposure. miR-29a-3p binds to TLR7 and elicits inflammation and causes neuronal damage.
Du et al. (2016) [113]	miR-127 miR-186 miR-222 miR-24 miR-329	Bcl2 PAK-LIMK2 p53	↑ miR-222, -24 ↓ Bcl2 ↑ miR-127, -24 ↓ PAK-LIMK2 ↑ miR-186, -24, -222, 127 ↓ miR-329 ↑ p53 ↓ Bcl2	The upregulation of miR-127, miR-186, miR-222, and miR-24 and downregulation of miR-329 affected synaptic plasticity of prefrontal cortex targeting genes involved in cell regulation of apoptosis and glialogenesis.
Sun et al. (2020) [114]	miR-181a	GluA2	↑ miR-181a ↓ GluA2	miR-181a may relate to the development of methamphetamine dependence with psychosis. miR-181a, binding within the miRNAs encoding the GluA2, can regulate synaptic function in hippocampal neurons.
Zhu et al. (2015) [115]	novel-m002C novel-m009C	Arc, Pde4c	↑ novel-m002C ↑ novel-m009C ↑ Arc ↓ Pde4c	The upregulation of novel-m002C and novel-m009 after exposure to methamphetamine in mice may affect gene expression of the pathway of addiction and brain plasticity.
Kim et al. 2022 [116]	miR-137	-	↓ miR-137	miR-137 in the circulating extracellular vesicles holds diagnostic potential as a blood-based marker of methamphetamine abstinence. However, the functional roles of circulating miR-137 in animals remain unexplored to date.
Zhao et al. (2016) [117]	miR-181a miR-15b miR-let-7e let-7d	-	↓ miR181a ↓ miR15b ↓ miR-let-7e ↓ miR-let-7d	The 4 aberrant expression miRNAs may be involved in the neuronal apoptosis. Negative correlation between the expression level of these miRNAs and drug exposure.
Demirel et al. (2023) [118]	let-7b-3p	MOR	↑ miR-let-7b-3p ↓ MOR	The upregulation of miR-let-7b-3p in drug users may decrease morphine and related μ-opioid receptor (MOR) expression.
Li et al. (2018) [119]	miR-128-3p miR-133a-3p miR-152-3p miR-181a-5p	SNARE cGMP-PKG GABAergic synapse	↑ miR-128-3p ↑ miR-133a-3p ↑ miR-152-3p ↑ miR-181a-5p ↓ SNARE ↓ cGMP-PKG ↓ GABAergic synapse	These miRNAs may have a strong association with drug addiction, and they may be involved in the different addiction processes, which partly explains methamphetamine addiction.
Shang et al. (2022) [120]	miR-222-3p	Ppp3r1 Cdkn1c Fmr1 PPARGC1A	↓ miR-222-3p ↑ Ppp3r1 ↑ Cdkn1c ↑ Fmr1 ↑ PPARGC1A	Putative targets of miR-222-3p are associated with cell apoptosis and synaptic plasticity and they are overexpressed during methamphetamine addiction.
Yang et al. (2020) [108]	miR-28-3p miR-547-3p miR-31 a-5p miR-3065-3p miR-338-5p miR-330-3p miR-3065-5p miR-338-3p 3560 miR-501-3p let-7b-3p miR-216a-3p miR-216b-5p novel 237 miR-217-5p miR-216b-miR-3p miR-31b miR-199a-5p miR-29b-3p miR-3587 miR-136-5p miR-153-3p miR-551b-3p miR-206-3p miR-486 miR-133a-miR-3p miR-133c miR-27a-5p miR-144-5p miR-1b miR-128-1-miR-5p miR-144-3p miR-200a-miR-3p miR-10a-5p miR-6315 novel 501 miR-542-3p novel 296 miR-202-5p miR-451-5p	-	↑ miR-28-3p ↑ miR-547-3p ↑ miR-31 a-5p ↑ miR-3065-3p ↑ miR-338-5p ↑ miR-330-3p ↑ miR-3065-5p ↑ miR-338-3p ↑ miR-3560 ↑ miR-501-3p ↑ let-7b-3p ↑ miR-216a-3p ↑ miR-216b-5p ↑ novel 237 ↑ miR-217-5p ↑ miR-216b-3p ↑ miR-31b ↓ miR-199a-5p ↓ miR-29b-3p ↓ miR-3587 ↓ miR-136-5p ↓ miR-153-3p ↓ miR-551b-3p ↓ miR-206-3p ↓ miR-486 ↓ miR-133a-3p ↓ miR-133c ↓ miR-27a-5p ↓ miR-144-5p ↓ miR-1b ↓ miR-128-1-5p ↓ miR-144-3p ↓ miR-200a-3p ↓ miR-10a-5p ↓ miR-6315 ↓ novel 501 ↓ miR-542-3p ↓ novel 296 ↓ miR-202-5p ↓ miR-451-5p	These miRNAs may contribute to methamphetamine-induced alterations in the nervous system through the Wnt signaling pathway, axon guidance, and lysosome pathways.
Sim et al. (2017) [121]	miR-496-3 miR-194-5p miR-200b-3p miR-181a-5p	DNMT3A GSTT2 PRTG GNAI3 TAOK1 ZC3H6 SSH2 USF3	↑ miR-496-3p - ↑ miR-194-5p ↓ DNMT3A ↓ GSTT2 ↑ miR-181a-5p ↓ PRTG ↑ GNAI3 ↓ miR-200b-3p ↑ TAOK1 ↑ ZC3H6 ↓ PRTG ↑ GNAI3 ↓ SSH2 ↓ USF3	Strong relationship between addiction biology and the genes that were differentially expressed. miR-496-3p, miR-194-5p, mir-200b-3p, and miR-181a-5p are highly altered in methamphetamine addiction and may be strongly associated with the addiction phenotype. Putative targets of these miRNAs are associated with cell apoptosis, cellular defense and neurogenesis.
Moradali et al. (2022) [122]	miR-127 miR-132	-	↓ miR-127 ↑ miR-132	The precise role of these miRNAs is still unknown in patients with methamphetamine abuse. Authors hypothesize a correlation with CREB, GnRH, and MAPK signaling pathways involved in synaptic transmission, angiogenesis, and inflammation.
Wang et al. (2021) [123]	miR-181a	ERAD GABAAα1	↓ miR-181a ↑ ERAD ↓ GABAAα1	Methamphetamine may activate the ERAD pathway through miR-181 and may induce the ubiquitination degradation of GABAAα1, which is involved in the formation of addiction.
Du et al. (2019) [124]	miR-143	PUMA NLRP3	↓ miR-143 ↑ PUMA ↑ NLRP3	Methamphetamine abuse induced NLRP3 inflammasome activation in microglia through MiR-143/PUMA axis.
Yu et al. (2019) [125]	miR-155-5p	Peli1	↓ miR-155-5p ↑ Peli1	The overexpression of miR-155-5p could directly suppress Peli1 expression and could protect against the inflammatory effects of methamphetamine treatment partially through activating p38 MAPK and NF-κB inflammatory pathways.
Liu et al. (2019) [126]	miR-124	Dicer1	↓ miR-124 ↑ Dicer1	Expression level of Dicer1, which is a potential target of methamphetamine-induced downregulated miR-124, is significantly increased indicating an active role of miRNAs in the development of drug addiction.
Su et al. (2019) [127]	miR-29c	Dnmt3a Dnmt3b Meg3	↑ miR-29c ↓ Dnmt3a ↓ Dnmt3b ↑ Meg3	miR-29c is an important epigenetic regulator of drug-induced behavioral sensitization and can interfere with gene expression (Dnmt3a, Dnmt3b, and Meg3).
Ni et al. (2019) [128]	miR-204-3p	Sema3A Plxna4	↑ miR-204-3p ↓ Sema3A ↓ Plxna4	Changes in Sema3A and Plxna4 are negatively correlated with miR-204-3p, which indicated that these genes involved in brain physiology and structural plasticity might be regulated by this miRNA in the expression of methamphetamine sensitization.
Liu et al. (2021) [129]	miR-3068-5p	Ago2 Grin1	↓ miR-3068-5p ↓ Ago2 ↑ Grin1	Neural Ago2/miR-3068-5p cascade is downregulated in mice during methamphetamine sensitization with overexpression of Grin1 involved in plasticity of synapses.
Sim et al. (2017) [121]	miR-9-5p	BDNF	↑ miR-9-5p ↑ BDMF	The highly altered expression of miR-9-5p in the chronic treatment group is related with the overexpression of the BDNF. This gene is involved in the dopamine pathway which may induce addictive disorders.
Bosch et al. (2015) [130]	miR-145 miR-129 miR-29	HDAC2 HDAC4	↓ miR-145 ↓ HDAC2 ↓ miR-129 ↓ HDAC2 ↓ miR-29 ↓ HDAC4	Downregulation of these miRNAs associated with differential regulation of multiple transcripts related to epigenetic mechanisms such as histone modification enzymes.
Shi et al. (2019) [131]	miR-134	LIMK1	↑ miR-134 ↓ LIMK1	miR-134 expression in Dorsolateral Striatum was significantly increased and its target LIMK1 expression was decreased in the excessive and uncontrolled methamphetamine self-administration rats.
Qian et al. (2021) [132]	miR-31-3p	RhoA	↑ miR-31-3p ↓ RhoA	Upregulation of miR-31-3p was associated with concomitant suppression of RhoA protein in the Dorsal Hippocampus of conditioned place preference mice induced by methamphetamine mice. RhoA is involved in the regulation of dendritic spine morphology, synaptic transmission, and synaptic plasticity.
Zhou et al. (2021) [133]	miR-183-5p	NRG1	↑ miR-183-5p ↓ NRG1	Upregulation of miR-183-5p suppresses NRG1, an epidermal growth factor (EGF)-like protein implicated in neural development and brain activity homeostasis.
Zhang et al. (2016) [134]	miR-181a	GRIA2	↓ miR-181a ↑ GRIA2	miR-181a significantly downregulated the expression of GRIA2 gene and inhibited the GRIA2 protein activity. This protein mediated the vast majority of excitatory neurotransmission in the brain and plays a crucial role in potentiating the functions of addictive drugs.
Kim et al. (2022) [135]	miR-137	SYNCRIP	↑ miR-137 (Dorsal Striatum) ↓ SYNCRIP ↓ miR-137 (circulating)	Aberrant increase in striatal miR-137-dependent inhibition of SYNCRIP essentially mediated the methamphetamine abstinence-induced reduction in circulating miR-137.

* Abbreviation: ↓ downespression ↑ upespression.

**Table 5 ijms-24-17122-t005:** miRNAs involved in Alcohol addiction and their effects.

Reference	Model	Tissue/Brain Region	Description of miRNA in Alcohol Abuse
Alhaddad et al. (2020) [31]	-	Hyppocampus	Increased levels of miR-124-3p and reduction of GRα and GRβ may have a contribution in alcohol addiction.
Tang et al. (2008) [33]	-	-	Upregulation of miR-212 disrupts tight junctions and raises cell permeability.
Gao et al. (2010) Zovoilis et al. (2011) [138,139]	-	NAc; Frontal cortex; PFC	miR34 and miR134 are upregulated in alcohol use disorder and act in the sretoninergic circuit, influencing learning and memory functions.
Qi et al. (2014) [36]	CGNs	-	miR29b’s greater degree of suppression occurs during the third and the ninth postnatal day.
Sathyan et al. (2007) [35]	Cortical-derived neurosphere cultures	-	miR21 and miR335 are suppressed between 7 and 9 h after alcohol transfection. miR21 and miR335 work as functional antagonists.
Darcq et al. (2015) [22]	HEK293 cells	-	miR1 is upregulated during binge drinking.
Lewohl et al. (2011) [18]	Humans	Frontal cortex	Up and downregulation of different patterns of miRNAs induce alcohol-related changes in the human brain that may be implicated in alcohol abuse.
Kim et al. (2007) [137]	Humans	Midbrain	Upregulation of miR133b influences maturation of midbrain dopaminergic neurons.
Tseng et al. (2019) [34]	Humans	Serum	miR140 was upregulated and released by fetal neural stem cells after 3 days of alcohol exposure.
Manzardo et al. (2013) [42]	Humans	Medial frontal cortex	Alcoholism induces the upregulation of miRNAs expressed in 14q32 chromosomic region. This modification is associated with a reduction in oligodendrocyte differentiation and with an increase in apoptosis.
Rosato et al. (2019) [48]	Humans	Saliva	Salivary miRNAs could be used as potentially biomarkers in order to predict alcohol dependence.
Coleman et al. (2017) [29]	Humans (postmortem) Rats	Hyppocampus Entorhinal cortex	Increased release of let-7b by microglia cells contributes to ethanol-induced neutoroxicity mediated by TLR7.
Santos-Bezerra et al. (2021) [41]	Humans (postmortem)	Hyppocampus	miR-34a and miR-34c were upregulated in subjects with AUD and may be involved in alcohol-related cognitive decline.
Asquith et al. (2014) [37]	Macaques	PBMC	MiR181 and miR221 were upregulated following chronic ethanol self-administration.
Most et al. (2016) [28]	Mice	mPFC	Levels of miR-411 are decreased after chronic consumption of alcohol.
Gorini et al. (2013) [140]	Mice	Cerebral cortex Midbrain	Different sets of miRNAs are expressed in cerebral cortex and in the midbrain. The expression of these sets varies according to transition from alcohol consumption to addiction.
Most et al. (2014) [44]	Mice	Synapses	Chronic alcohol consumption perturbs miRNAs expression in synapses and, consequently, mRNA and proteins expression. Such effect may determine alterations in synaptic plasticity.
Nunez et al. (2013) [141]	Mice	Frontal cortex	Early stages of alcohol dependence, overexpression of miRNAs is not accompanied by downregulation of their own targets.
Bekdash et al. (2015) [38]	Murine cortical neurons	-	After alcohol withdrawal between 3 and 24 h, Gabra4 levels were downregulated and miR186, miR24, miR375, and miR155 were upregulated.
Guo et al. (2012) [40]	Murine Cortical neurons	-	Upregulation of miR-155, -199a-3p, and -685, and reduction of MeCP2 may contribute to the mechanism of withdrawal-induced hyperexcitability.
Balaraman et al. (2012) [30]	Ovine	Plasma	Ethanol exposure determines miRNA modifications that may induce fetal developmental defects. Prenatal alcohol exposure changes miRNA profiles in two-week and six-months infants.
Ehinger et al. (2020) [19]	Rats	DLS; mPFC; serum	miR30a-5p, miR206, miR191, and miR195 are upregulated after heavy alcohol intake.
Tapocik et al. (2013) [20]	Rats	mPFC	miR206 was upregulated after 3 weeks of withdrawal.
Bahi et al. (2013) Mizuo et al. (2016) [21,26]	Rats	DLS; whole brain	miR124a is downregulated after voluntary alcohol intake. Its levels are, however, elevated in abstinence.
Darcq et al. (2015) #[22]	Rats Humans	mPFC	miR30a-5p and miR195-5p are upregulated in binge drinking and after 24 h of abstinence. Upregulation of miR30a-5p is related to an increased alcohol consumption.
Pietrzykowski (2012) Gu et al. (2018) Li et al. (2011) [23,24,25]	Rats Neurons Humans	Supraoptic nucleus; Striatum; Hypophysis	miR9 is upregulated after an acute intake of alcohol and contributes to the tolerance of the substance. It also determines reduction of levels of HDA5, implicated in behavioral responses to chronic drug exposure. Moreover, it is involved in the induction of alcohol addiction.
Li et al. (2013) [27]	Rats	NAc	When overexpressed, mir-382 inhibits DR1 expression and reduces alcohol consumption.
Bahi et al. (2013) [21]	Rats	NAc	When overexpressed, let-7d inhibits DR3 expression and reduces alcohol consumption.
Choi et al. (2020) [39]	Rats	Hyppocampus	Alcohol is able to produce sex-dependent effects on neurogenesis and regulation of miRNAs.

**Table 6 ijms-24-17122-t006:** miRNAs involved in Cocaine addiction and their effects.

Reference	Model	Tissue/Brain Region	Description of miRNA in Cocaine Abuse
Barreto-Valer et al. (2012) [50]	Zebrafish embryos	-	Downregulation of miR-133b at 24 and 48 hpf.
Bastle et al. (2018) [51]	C57BL/6J mice and Sprague Dawley rats	Brain	miR-495 significantly downregulated at 1 and 4 h post injection.
Cabana-Dominguez et al. (2018) [52]	SH-SY5Y cells	-	miR-9, miR-153, and miR-124 downregulated after acute exposure to cocaine.
Chandrasekar et al. (2009) [53]	Male Wistar rats	Brain	miR-124 and let-7d downregulated whilst miR-181 was overexpressed.
Chandrasekar et al. (2011) [54]	Male Wistar rats	Brain	miR-124 and let-7d downregulated whilst miR-181 was upregulated.
Chen et al. (2013) [55]	Wistar rats	Brain	25 miRNAs were upregulated after 14 days of exposure to cocaine, while 9 were downregulated. At day 26, after a 12-day extinction period, 9 miRNAs were downregulated whilst 33 were overexpressed.
Chivero et al. (2020) [56]	C57BL/6 mice	Brain	Downregulation of miR-124 after 7 days cocaine exposure.
Dash et al. (2017) [57]	SH-SY5Y cells	-	Downregulation of miR-125b after cocaine exposure.
Dash et al. (2020) [58]	SH-SY5Y cells and C56BL/6 mice	Brain	miR-124 was downregulated after cocaine exposure.
Doke et al. (2021) [59]	Human	Astrocytes	Upregulation of miR-135-2 and miR428e, while miR5580 and miR6799 were significantly downregulated.
Doke et al. (2022) [60]	Human	Astrocytes	Hsa-miR-2355 and hsa-miR-4726-5p downregulated in cocaine exposed astrocytes.
Domingo-Rodriguez et al. (2021) [61]	Male C57BL/6 mice	Brain	Mmu-miR-34b-5p and mmu-miR-1249-3p downregulated in NAc in vulnerable or resilient animals to cocaine addiction-like behavior.
Dykxhoorn et al. (2023) [62]	Male C57BL/6 mice	Aorta	miR-30c-5p upregulated whilst miR-423-5p was downregulated after cocaine exposure.
Forget et al. (2022) [63]	Mice	Striatum	Upregulation of miR-1, miR-16, miR-29b, miR-31, miR-32, miR124, miR-125b, miR-128b, miR-132, miR-181a, miR-212, miR-221, miR-223, and Let7d in the NAc after 10 days exposure.
Gawlinski et al. (2022) [64]	Wistar rats	Striatum and hippocampus	Downregulation of miR-374 and miR-544 in the striatum and hippocampus after 14 days of self-administering cocaine.
Giannotti et al. (2014) [65]	Sprague Dawley rats	Medial prefrontal cortex	Downregulation of let7-d, miR-124, and miR-132.
Guo et al. (2016) [66]	BV-2 cells and Sprague Dawley rats	Microglia	Cocaine exposure decreased miR-125 levels in BV-2 cells and in rats microglia in vivo.
Hollander et al. (2010) [67]	Male Wistar Rats	Dorsal striatum brain region	miR-212 and miR-132 upregulated after 6 h of intravenous cocaine self-administration.
Im et al. (2010) [68]	Rats	Dorsal striatum brain region	miR-212 upregulated after cocaine exposure.
Kumaresan et al. (2023) [69]	Rats	Brain	Acute 18 h withdrawal resulted in the upregulation of rno-miR-338-3p in the PL and its downregulation in the NAc; the downregulation of rno-miR-382-3p in the IL and NAc. After a 4 weeks withdrawal period, rno-miR-935 and rno-miR-770-5p were upregulated in NAc and IL.
Li et al. (2020) [70]	C57BL/6 male mice	Brain	miR-134-5p up regulated during cocaine extinction period.
Lòpez-Bellido et al. (2012) [71]	Zebrafish embryos	-	Upregulation of let-7d at 24 and 48 hpf.
Mantri et al. (2012) [72]	PBMC	-	Downregulation of miR-125b, miR-150, miR-28-5p, miR-233, and miR-382 in cocaine treated cells.
Napuri et al. (2013) [73]	MDDC cells	-	miR-155 and miR-20a downregulated after cocaine exposure.
Nudelman et al. (2009) [74]	Male CB57L/6 mice	Brain	miR-132 overexpressed at 10, 17, and 24 days after birth. Expression increased significantly with age.
Periyasamy et al. (2018) [75]	BV-2 and HEK293 cells. Sprague Dawley rats and C57BL/6N mice.	Microglia	miR-124 downregulated in BV-2 cells and rPMs after cocaine exposure. Cocaine reduces expression of miR-124 in mice brains.
Quinn et al. (2015) [76]	Male Sprague Dawley rats	Striatal subregions of the brain	miR-431 was overexpressed in addiction-vulnerable rats compared to resilient ones. No differences in miR-221 expression.
Quinn et al. (2018) [77]	Male Sprague Dawley rats	Striatal subregions of the brain	Downregulation of miR-101b, miR-137, and miR-132 in cocaine self-administration group. In addiction-prone rats, miR-212 is overexpressed early in the addiction cycle. MiR-101b, miR-137, miR-212, and miR-132 upregulated late in the addiction cycle.
Sadakierska-Chudy et al. (2017) [78]	Male Wistar rats	Striatum	Up regulation of miR-132 and miR-212 following cocaine in self-administration group. After 14 days of intake, no upregulation was found in both yoked saline and yoked cocaine groups.
Schaefer et al. (2010) [79]	FLAG-Ago2 mice and Drd2 neurons	Striatum	miR-467, miR-544, miR-154, miR-467b, miR-500, miR-186, miR-337-3p, miR-138, miR-369-3p, miR-136, miR-7a, miR-137, miR-324-5p, miR-665, miR-380-3p, miR-376a, miR-130a, miR-301b, miR-148b, miR-488, miR-384-5p, miR-376c, and miR-181a were upregulated in Drd2 neurons and downregulated in Ago2 deficient mice by acute cocaine use.
Vannan et al. (2021) [80]	Male Sprague Dawley rats	Nucleus accumbens	8 miRNAs were downregulated and 25 were upregulated in low-seeking mice group relative to the high-seeking ones. 8 of the 33 miRNAs identified has correlated with high-seeking behavior, 5 positively and 3 negatively
Tobon et al. (2015) [81]	Mice	Caudate/putamen brain region	Down regulation of miR142-3p and miR382 from 5 min to 15 min after cocaine challenge in cocaine-sensitized mice
Viola et al. (2016) [82]	Male BALB/6 mice	Frontal areas of the brain	Down regulation of miR-212 2 h after cocaine induced CPP test.
Viola et al. (2019) [83]	Human	Peripheral blood	Up regulation of miR-124 and miR-181 in cocaine users, no difference in miR-212 between cocaine users and the control group.
Xu et al. (2013) [84]	Rats	Dorsal striatum brain region	miR-212 up regulated at 1 h and 6 h after cocaine exposure.
Zhu et al. (2018) [85]	Mice	Aortic vessels	miR-30c-5p up regulated in cocaine exposed mice while Me1 is significantly reduced.

**Table 7 ijms-24-17122-t007:** miRNAs involved in Opioid addiction and their effects.

Reference	Model	Tissue/Brain Region	Description of miRNA in Heroine Abuse
Sanchez-Simon et al. (2010) [107]	Zebrafish Hippocampal neurons (murine)	-	Morphine downregulated the miR-133b. Such an effect was abolished by the coadministration of morphine and naloxone.
Jia et al. (2022) [88]	Rats	Dentate gyrus neurons	miR-132 is upregulated following the administration of increasing doses of morphine twice a day for six consecutive days and a single injection of naloxone at 6 h after the last morphine injection.
Zhao et al. (2017) [96]	Rats	mPFC	Overexpression of miR-105 in the mPFC leads to suppression of D1 receptor expression in glutamatergic terminals of the projection neurons from the mPFC to the BLA, and a reduction in CPA in morphine withdrawal.
Wu et al. (2008) [102]	NS20Y (murine neuroblastoma cells) #COS-1 (monkey kidney cells)	-	MiR23b interacts with the MOR1 3′-UTR complex, inhibiting expression of MOR1.
Wu et al. (2009) [98]	N2A and N2A-MOR cells (mice) NMB and SHSY-5Y (human neuronal cells)	Neuro and brain cell	Morphine inhibits the association of MOR1 mRNA with polysomes through an interaction between miRNA23b and MOR1 3′-UTR.
Liao et al. (2020) [92]	Mice A172 cells (human astrocytes)	Astrocytes/microglia	Micro-Rna-138 is upregulated and it is implicated in neuroinflammatory response.
Xie et al. (2022) [94]	Mice	Nucleus accumbens	miR-592-3p regulated expression of TMEFF1 in NAc during cue-induced incubation of morphine craving.
Kim et al. (2018) [97]	Mice	Nucleus accumbens	33 miRNAs were upregulated and 29 downregulated in mice treated with morphine.
Zheng et al. (2010) [103,104,105]	Mice Hippocampal neurons (murine) Hippocampal neurons (murine) Hippocampal neurons (murine)	Hippocampus	Fentanyl, but not morphine, downregulates miR-190 after 24 h of treatment. The effect persisted for 72 h. Downregulation is blocked by naloxone, U0126.
He et al. (2010) [106]	Mice SH-SY5Y cells	-	Chronic treatment with morphine significantly upregulated let-7 expression, that might be correlated to development of opioids tolerance.
Michelhaugh et al. (2011) [101]	Human (postmortem)	Nucleus accumbens	These lncRNA are upregulated in heroine abusers.
Gu et al. (2020) [89]	Human	Serum	The serum levels of miRNAs were significantly increased in heroin abusers.
Liu et al. (2021) [90]	Human	Blood	The serum levels of these microRNA are upregulated in heroin dependence.
Wenjin et al. (2020) [91]	Human	Plasma	Micro-RNA 181a is upregulated.
Hsu et al. (2019) [93]	Human	Blood	miR-133b downregulates the homeodomain transcription factor Pitx3. miR-133b levels in subjects with OUD were lower than that in healthy controls.
Dave and Khalili (2010) [100]	Human	Blood	Morphine induced the expression of miR-15b and decreased the expression of miR-18b.
Yan et al. (2017) [95]	-	Nucleus accumbens	Mir-218 was found downregulated after chronic intake of heroin.

**Table 8 ijms-24-17122-t008:** miRNAs involved in Methamphetamine addiction and their effects.

Reference	Model	Tissue/Brain Region	Description of miRNA in Methamphetamine Abuse
Sim et al. (2017) [121]	Rats—Male Wistar	Nucleus Accumbens	Upregulation of miR-496-3p, miR-194-5p, and miR-181a-5p and downregulation of miR-200b-3p were found to be significantly associated with methamphetamine addiction.
Sim et al. (2017) [121]	Rats—Male Wistar	Nucleus Accumbens	Upregulation of miR-9-5p in the acute and chronic methamphetamine treatment group.
Du et al. (2016) [113]	Rats—Male Sprague Dawley	Prefrontal Cortex	miR-127, miR-186, miR-222, and miR-24 were upregulated and miR-329 was downregulated in escalated methamphetamine self-administration rats compared with the control group.
Li et al. (2018) [119]	Rats—Male Sprague Dawley	Serum exosomes	Upregulation of miR-128-3p, miR-133a-3p, miR-152-3p, and miR-181a-5p in methamphetamine dependent rats.
Yang et al. (2020) [108]	Rats—Male Sprague Dawley	Nucleus Accumbens	A total of 40 differentially transcribed miRNAs in comparison with the control group were found: 17 upregulated and 23 downregulated by methamphetamine treatment.
Wang et al. (2021) [123]	Rats—Male Sprague Dawley	Dorsal Striatum	Downregulation of miR-181 in the dorsal striatum of methamphetamine-addicted rats.
Bosch et al. (2015) [130]	Rats—Male Sprague Dawley	Ventral Tegmental Area	Downregulation of miR-145, miR-129 and miR-29 in methamphetamine self-administration rats.
Shi et al. (2019) [131]	Rats—Male Sprague Dawley	Dorsolateral Striatum	Upregulation of miR-134 in excessive and uncontrolled methamphetamine self-administration rats.
Du et al. (2019) [124]	Mice Male BV2 cells	Microglia	Downregulation of miR-143.
Yu et al. (2019) [125]	Mice Male BV2 microglial cells	Cortical Tissues	Downregulation of miR-155-5 associated with overexpression of genes involved in inflammatory effects.
Liu et al. (2019) [126]	Mice (Male)	Nucleus Accumbens	Downregulation of miR-124 in methamphetamine-sensitized mice.
Su et al. (2019) [127]	Mice (Male)	Nucleus Accumbens	Upregulation of miR-29c in acute methamphetamine-treated mice.
Ni et al. (2019) [128]	Mice (Male)	Nucleus Accumbens	Upregulation of miR-204-3p in the nucleus accumbens in methamphetamine-sensitized mice.
Liu et al. (2021) [90]	Mice (Male)	Nucleus Accumbens	Downregulation of miR-3068-5p in the nucleus accumbens during methamphetamine-sensitization.
Qian et al. (2021) [132]	Mice (Male)	Dorsal Hippocampus	Upregulation of miR-31-3p in conditioned place preference mice induced by methamphetamine.
Zhou et al. (2021) [133]	Mice (Male) H9c2 cardiomyoblast HT22 neuronal precursor	Hippocampus	Upregulation of miR-183-5p in conditioned place preference mice induced by methamphetamine.
Shang et al. (2022) [120]	Mice	Nucleus Accumbens	Downregulation of miR-222-3p in methamphetamine-induced conditioned place preference mice.
Zhu et al. (2015) [115]	Male mice	Nucleus Accumbens	novel-m002C and novel-m009C were upregulated in mice exposed to methamphetamine.
Chand et al. (2021) [112]	Macaque Rhesus Sprague Dawley rats	Frontal grey neurons and microglia	miR-29a-3p is upregulated in chronic methamphetamine exposure (28 weeks) leading to neuroinflammation and neuronal damage.
Sun et al. (2020) [114]	Human methamphetamine users	Hippocampus	miR-181a expression was increased in methamphetamine use disorder with psychosis.
Demirel et al. (2023) [118]	Human (postmortem)	Nucleus Accumbens Ventral Tegmental Area	Upregulation of miR-let-7b-3p in methamphetamine users who have died by methamphetamine acute intoxication.
Kim et al. 2022 [116]	Human	Circulating extracellular vesicles (cEVs)	miR-137 is downregulated in patients under methamphetamine abstinence.
Zhao et al. (2016) [117]	Human	Plasma	Downregulation of miR181a, miR15b, miR- let-7e, and miR-let-7d in methamphetamine users.
Moradali et al. (2022) [122]	Human	Peripheral Blood	Downregulation of miR-127 and upregulation of miR-132 in patients with methamphetamine abuse compared with the control group.
Zhang et al. (2016) [134]	Human	Serum	Chronic methamphetamine use downregulates the expression of miR-181a in human serum.
Kim et al. (2022) [135]	Female Macaca fascicularis Male mice	Dorsal Striatum Blood	Methamphetamine abstinence triggers upregulation of miR-137 in Dorsal Striatum and downregulation of the same miRNA in blood due to the feedback of RNA-binding protein SYNCRIP.

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
