# Peer review of "miRNAs and Substances Abuse: Clinical and Forensic Pathological Implications: A Systematic Review"

_ijms, 2023, doi:10.3390/ijms242317122_

Round 1

Reviewer 1 Report

Comments and Suggestions for Authors

The authors review the substances of abuse and the miRNAs involved in the pathophysiology.

The authors should address some minor issues:

1. The authors should explain more clearly in their methodology the PICO model and the variables used.

2. Assessment of bias the results obtained should be indicated.

3. line 110. The authors indicate 132 scientific papers included and fig. 1 show 136??

4. Could the authors indicate in their discussion if there are miRNAs common to all substances of abuse and their possible mechanisms of action.

5. It would be interesting in each table to group references on studies in humans and other animals.

Author Response

We appreciate your valuable contributions in improving this paper. Enclosed, you'll discover the integration of your suggestions into our work: please see the attachment. 

Reviewer 2 Report

Comments and Suggestions for Authors

This work provides an overview of current literature on miRNAs involved in addic- 24 tion, evaluating their impaired expression and regulatory role in neuroadaptation and synaptic 25 plasticity

Overall this is an interesting paper, but a major revision is needed.

The introduction needs more references to support statements. The same applies in the discussion. In addition, the cross-cultural differences not only for the addicted person, but also for their environment regarding addiction should be briefly discussed. Biological addictions in humans appear in a socio-cultural environment that may differ even when we discuss about individuals in western european countries and this environement (e.g. created by the family members or other significant others) may change perception and expression of addiction (e.g. Giannouli, V., & Ivanova, D. (2016). Codependency in mothers of addicted persons: Cross-cultural differences between Greece and Bulgaria. European Psychiatry, 33(S1), S622-S623.).

The methodology section needs rewriting. There is not enough information presented to the reader (although the figure is ok). Describe in more detail how were the articles found, when, by whom, what databases were used.

The same applies to the results section. More details coming from the Table must be discussed.

Comments on the Quality of English Language

Minor English language editing is needed.

Author Response

Dear Reviewer 2:

Our Research team is grateful to Reviewer 2 for his attention and advises. We shared the comments provided by reviewer 2 with our research team. They highlighted several areas for improvement, including introduction, and a revised interpretation of the 'methods' paragraph; also, is stressed tables contents in result. 

 Here, you will find the changes of how we have implemented the work based on your suggestions. 

Thanks to your suggestion introduction is implemented discussing the social factors that predict drug abuse. Also, themes and role of codependency in drug abuse  are highlighted by expanding the bibliography. 

Following the guidance provided by the reviewer, we have modified the material and methods section. We have provided a more detailed description of the systematic literature review process:

"2.2 Search Criteria and Critical Appraisal A systematic literature search and a critical appraisal of the collected studies were conducted. An electronic search of PubMed, Science Direct Scopus, and Excerpta Medica Database (EMBASE) from the inception of these databases to September 2023 was performed. The search terms were: “miRNA expression in addiction”; “miRNAs during alcohol addiction”; “miRNAs during cocaine addiction”; “miRNAs during methamphetamine addiction”; “miRNAs during opioids addiction” in the title, abstract, and keywords. The bibliographies of all identified documents were reviewed and cross-checked for other relevant literature. Methodological assessment of each study was conducted according to PRISMA standards, including assessment of bias. Data collection included study selection and data extraction. Three researchers (C.O., J.L, N.I.) independently reviewed those documents whose title or abstract seemed relevant and selected those that analysed miRNAs. The question of the suitability of miRNAs was resolved by consensus among the researchers. Unpublished or grey literature was not searched. Data extraction was performed by three investigators (E.T., A.M., A.C.) and reviewed by two other researchers (V.F., P.F.). Only English-language papers or abstracts were included in the search.

2.3 Search Results and Included Studies An appraisal based on titles and abstracts, as well as a hand search of reference lists, were carried out. The reference lists of all located articles were reviewed to detect still unidentified literature. The resulting reference lists were then screened for title and abstract, leaving 216 articles, for further consideration, that were then screened based on their abstract to identify their relevance in respect to the following: Estimate diagnosis process; Clinical features analyzed; Circumstantial data evaluation; Study design. The methodology of our search strategy is represented in Figure 1. A further categorization of the articles included in the study was made based on the main parameter under study, as showed by the following table (Figure 1). Non-English articles were excluded. The inclusion criteria      were as follows: (1) original research articles, (2) case reports/series. Reviews and meta-analyses were excluded. Studies conducted on cultured cells, animals (mainly rats and mice) and humans that described the role of miRNAs in different stages of addiction on four types of substances of abuse (alcohol, cocaine, opioids and methamphetamine) were selected. Among them, studies measuring miRNAs in CNS and some body fluids (mainly blood and saliva) from animals and humans were included, as also studies with a control group. These publications were carefully assessed, considering the main objectives of the review. After this evaluation, 132 scientific papers remain 

2.3 Risk of bias This systematic review has several strengths that include the amount and breadth of the studies, which span the globe; the hand search and scan of reference lists for the identification of all relevant and significant studies; and a flowchart that describes in detail the study selection process. Included studies were evaluated according to the Quality Assessment of Diagnostic Accuracy Studies 2 (QUADAS-2) tool to assess the risk of bias. This review includes studies that were published in a time frame of 13 years (from 2010 to 2023); thus, despite our efforts to fairly evaluate the existing literature, study results should be interpreted taking into account that the accuracy of the clinical procedures, here reported, has changed over the years. "

According to reviewer request, we discuss article-by-article any findings in the results section, then we summarized them in tables. We have completed a thorough assessment of the English and grammar.

We extend our gratitude for your valuable input in enhancing this paper. In red, you will find how we have incorporated reviewers' suggestions into our work.